# The Augmented Tree Tensor Network Cookbook

**Nora Reinić**[1,2], **Luka Pavešić**[1,2] **Daniel Jaschke**[3,2,1] **and Simone Montangero**[1,2]

**1** Dipartimento di Fisica e Astronomia "G. Galilei" & Padua Quantum Technologies Research Center, Università degli Studi di Padova, Italy I-35131, Padova, Italy
**2** INFN, Sezione di Padova, via Marzolo 8, I-35131, Padova, Italy
**3** Institute for Complex Quantum Systems & Center for Integrated Quantum Science and Technology, Ulm University, Albert-Einstein-Allee 11, 89069 Ulm, Germany

## Abstract

An augmented tree tensor network (aTTN) is a tensor network ansatz constructed by applying a layer of unitary disentanglers to a tree tensor network. The disentanglers absorb a part of the system's entanglement. This makes aTTNs suitable for simulating higher-dimensional lattices, where the entanglement increases with the lattice size even for states that obey the area law. These lecture notes serve as a detailed guide for implementing the aTTN algorithms. We present a variational algorithm for ground state search and discuss the measurement of observables, and offer an open-source implementation within the Quantum TEA library. We benchmark the performance of the ground state search for different parameters and hyperparameters in the square lattice quantum Ising model and the triangular lattice Heisenberg model for up to $32 \times 32$ spins. The benchmarks identify the regimes where the aTTNs offer advantages in accuracy relative to computational cost compared to matrix product states and tree tensor networks.

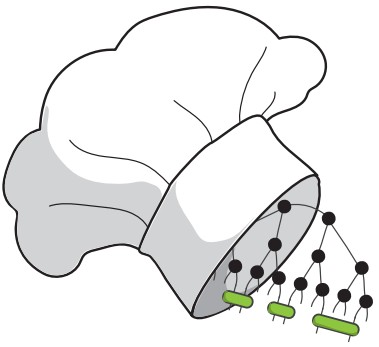

# 1 Introduction

Tensor network methods are a family of widely-used numerical methods developed for simulating quantum many-body systems [1–3]. A tensor network is an ansatz for a quantum state, constructed such that the computational complexity of the underlying operations and algorithms scales polynomially with the bond dimension $m$. The bond dimension – the size of the tensors in the network – corresponds to the number of Schmidt coefficients of the state's bipartitions and therefore determines the amount of entanglement that the tensor network

can faithfully capture. This typically also increases with the complexity of the network. Simultaneously, the polynomial degree of the algorithm's computational complexity depends on the details of the various ansätze. One thus aims to find a balance between how well an ansatz captures the state's physical properties and how numerically efficient or inefficient the corresponding algorithms are.

The most successful ansatz for one-dimensional systems is the matrix product state (MPS) [4–6], where the tensors are arranged in a line. The cost of the variational ground state search for the MPS [7] scales as $\mathcal{O}(m^3)$. The MPS is capable of representing the quantum states that obey the area law of entanglement in one physical dimension. This property is important in one dimension because the ground states of local gapped Hamiltonians often obey the area law of entanglement [8–10]. Other common ansätze are the projected entangled-pair state (PEPS) [11–13] – the two-dimensional analogue of the MPS – and the multi-scale entanglement renormalization ansatz (MERA) [14], designed for simulating one-dimensional systems at criticality [15]. However, manipulating PEPS or MERA is numerically challenging due to their loopy structure. This leads to high computational cost of the underlying algorithms (for PEPS optimization commonly scaling as $\mathcal{O}(m^{10})$ [16], improving to $\mathcal{O}(m^9)$ with a sampling-based approach in recently introduced methods [17], and $\mathcal{O}(m^{\geq 7})$ for MERA optimization in one dimension [18] or $\mathcal{O}(m^{16})$ in two dimensions [19]). Another possibility is given by tree tensor networks (TTN) [20,21], which represent the state with a tree-like structure. The TTN is a middle ground between the MPS and PEPS/MERA. It provides more adaptability to higher-dimensional lattices and long-range interactions compared to the MPS, while keeping a reasonably efficient optimization algorithm, $\mathcal{O}(m^4)$ [22] for one-tensor updates. However, the TTN architecture does not capture the area law in two dimensions [23,24].

Recently, another pathway has been introduced in Refs. [25–27] with the augmented tree tensor network (aTTN) ansatz, whose geometry enables capturing the entanglement area law in any dimension [25]. The aTTN is augmented with a layer of disentanglers applied to the physical links of a TTN. It represents a subclass of MERA, with disentanglers applied only on the lowest layer. So far, a ground state search optimization algorithm was introduced for the aTTNs. However, identifying the regimes where aTTNs give the advantage in terms of precision versus computational resources with respect to other tensor network ansätze is not straightforward.

This cookbook aims to be a detailed and practical guide through the aTTN operations, giving the recipes for implementation of the ground state search optimization and measurement of observables, and providing insights into the performance of the optimization algorithm in various parameter and hyperparameter regimes. All the described algorithms are implemented within the open-source tensor network library Quantum TEA [28]. We also provide pedagogical Jupyter Notebooks for running the aTTN ground state search in the supplemental material [29]. Therefore, the cookbook is intended for everyone interested in understanding and implementing the algorithm themselves, or using it with the Quantum TEA library. We proceed with the assumption that the reader is familiar with standard tensor network concepts. Otherwise, see Refs. [1,2,7,30,31].

The cookbook is structured as follows. First, an introduction to the aTTN geometry is given in Sec. 2. We define and set the notation for all relevant tensor network objects in Sec. 3. Then, the ground state search algorithms are described in Sec. 4, with a detailed explanation of two different approaches to the disentangler optimization. We then proceed to the description of the measurement of the observables in Sec. 5, highlighting the differences compared to the TTN implementation. Finally, we give an insight into how the aTTNs perform in different regimes and in which cases we can expect the advantage in comparison to the MPS and TTN in Sec. 6. We show the results of the benchmarks using GPU on the square lattice quantum Ising model and triangular lattice Heisenberg model, for different combinations of aTTN parameters

and hyperparameters. We conclude with a summary and an outlook in Sec. 7.

## 2    What is an augmented tree tensor network?

The augmented Tree Tensor Network (aTTN) is a tensor network ansatz defined as:

$$|\psi_{\mathrm{aTTN}}\rangle = D(u)|\psi_{\mathrm{TTN}}\rangle, \tag{1}$$

where $|\psi_{\mathrm{TTN}}\rangle$ is a Tree Tensor Network (TTN), and $D(u) = \prod_k u_k$ is a set of two-site unitary gates $u_k$, called disentanglers, applied on the TTN's bottom layer, i.e., tensors with physical links. The graphical representation of the aTTN for an example with three disentangler gates is shown in Fig. 1(a).

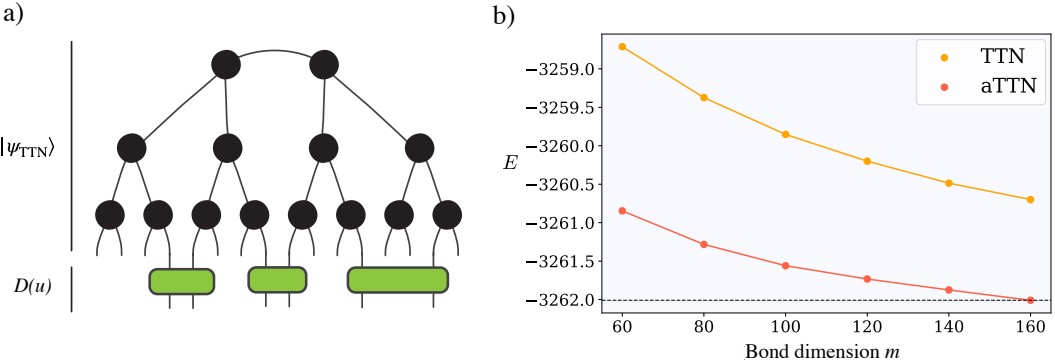

Figure 1: **The augmented tree tensor network (aTTN) ansatz.** (a) An aTTN with three disentanglers for a system of $N = 16$ sites. The full aTTN state $|\psi_{\mathrm{aTTN}}\rangle$ consists of a tree tensor network state $|\psi_{\mathrm{TTN}}\rangle$ (black circles) and a disentangler layer $D(u)$ (green rectangles) applied to $|\psi_{\mathrm{TTN}}\rangle$. (b) Comparison between the TTN and the aTTN ground state search energies for different bond dimensions $m$, computed for the nearest-neighbour quantum Ising model on a $32 \times 32$ lattice near the critical point. The aTTN shows an improvement in energy with respect to the TTN for every bond dimension point.

The role of the disentanglers is to encode the entanglement between the subsystems they connect. This allows the aTTN ansatz to represent states with more entanglement compared to the TTN with the same bond dimension. As a demonstration, in Fig. 1(b) we compare the ground state energies obtained with the TTN and the aTTN for the nearest-neighbour quantum Ising Hamiltonian $\hat{H} = -\sum_{<ij>} \sigma_x^i \sigma_x^j - h\sum_i \sigma_z^i$ on a $32 \times 32$ lattice with open boundaries. The energies are computed for $h = 3$, close to the critical point. We can see that the aTTN results in lower energies with respect to the TTN for all bond dimensions $m$, i.e., the ground state is represented more accurately.

    With the ability to contain entanglement in the disentangler layer, the aTTNs are capable of encoding the area law of entanglement in any number of dimensions [25], given that enough disentanglers are placed over certain bonds. Nevertheless, increasing the number of disentanglers increases the computational complexity of the aTTN algorithm, as we discuss in Sec. 4.4.

    The question at hand is: how does one find the optimal set of disentanglers such that the aTTN accurately represents the target state? So far, the algorithm for finding the ground state of a given Hamiltonian has been developed. The algorithm details are presented and

explained in the following sections. Note that we are following the procedure introduced in Refs. [25, 26]. An alternative approach was presented in the Ref. [27].

# 3  Ingredients

Before proceeding with the aTTN ground state search algorithm, we provide a short revision and set the notation for the relevant tensor network objects. In particular, we work with TTNs, disentanglers, and matrix product operators (MPOs), which represent the Hamiltonian terms, and the effective operators. For a summary, see the cheat sheet in Fig. 2. For a more detailed description, see Secs. 3.1-3.3.

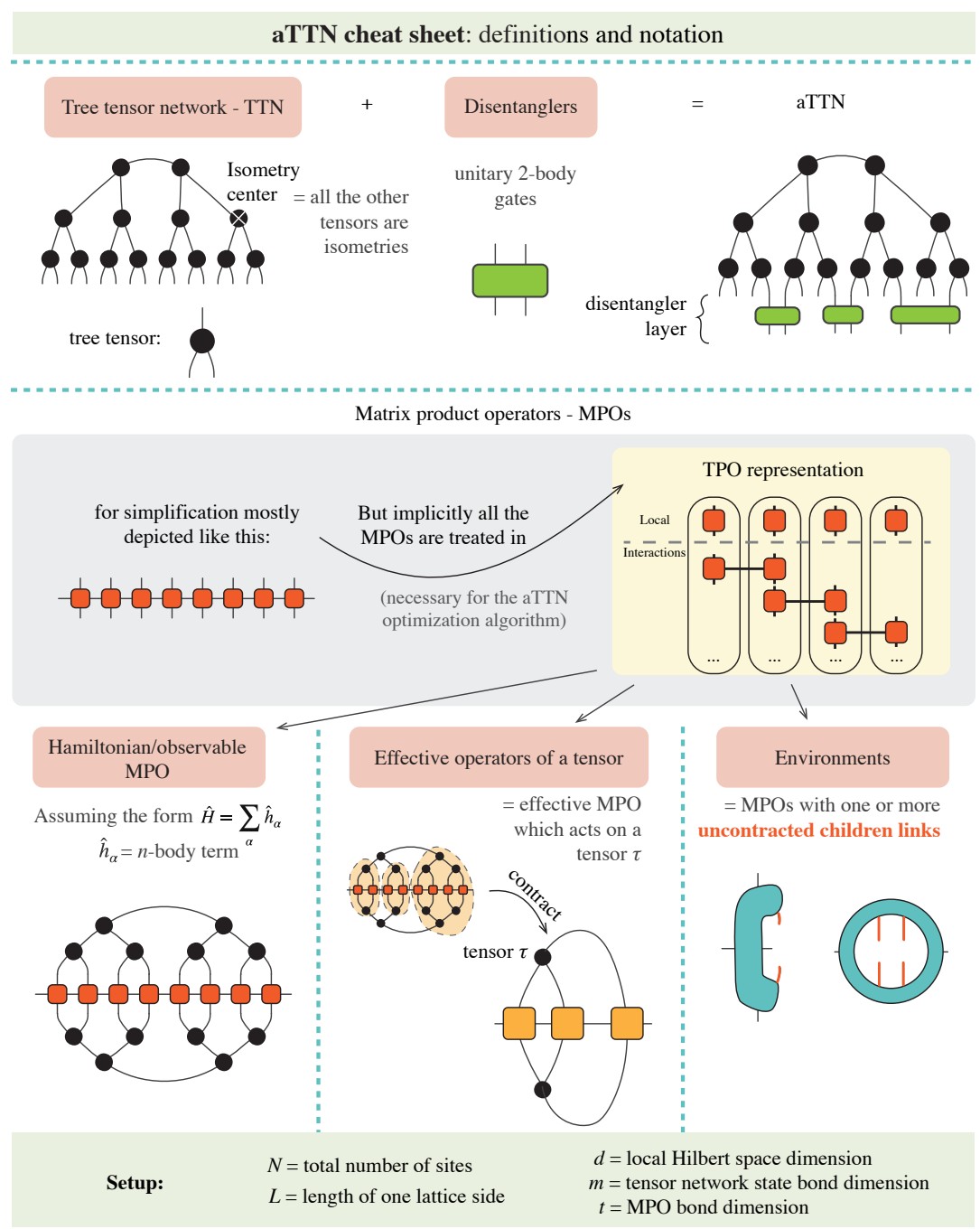

Figure 2: **Notation and ingredients for the augmented tree tensor network (aTTN) cookbook.** The colors of the tensor network objects depicted here are used consistently in all subsequent figures. Here and throughout, the dashed boxes denote the contraction of tensors inside them.

## 3.1  Tree tensor network

A binary TTN ansatz decomposes a wave function into a binary tree structure, as shown with the black rank-three tensors in Fig. 1(a). Hereafter, we refer to the tensor's links connected to the lower layer as the children links, and to the links connected to the upper layer as the parent links, for a TTN orientation where the bottom layer corresponds to physical sites. The total number of layers for an $N$-site system is $\log_2(N) - 1$ and each of the open links in the

lowest-layer tensors corresponds to a physical site.

A tensor $T$ is an isometry over a set of links if contracting $T$ with $T^\dagger$ over the remaining links yields an identity tensor (Fig. 3).

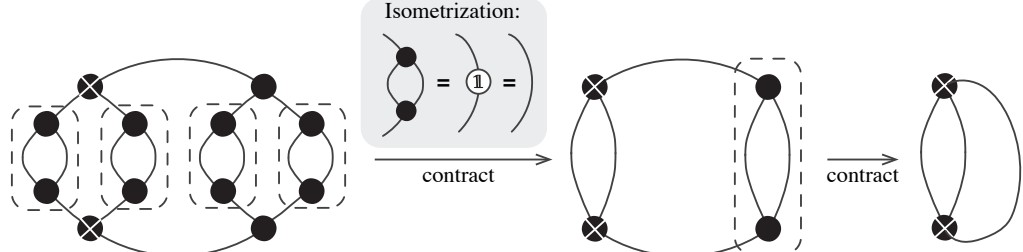

Figure 3: **Example of an isometry tensor.** When contracted with its Hermitian conjugate over the specific links, the isometry tensor $T$ yields an identity.

Every TTN can be isometrized using QR-decompositions such that all the tensors apart from one are isometries if contracted with their complex conjugates over the right indices. The only non-isometry tensor in a network is called the isometry center. Isometrizing a tensor network greatly simplifies the operations in general, for example, when computing the expectation values. An example is demonstrated in Fig. 4 for the 2-norm computation, showing that computing the norm of an entire TTN corresponds to computing the norm of only the isometry center tensor. From now on, we mark a tensor with a white cross whenever the isometry center is labeled explicitly in the tensor network.

Figure 4: **Computing the 2-norm of an isometrized tree tensor network (TTN) reduces to computing the 2-norm of the isometry center tensor.** An example is depicted for an eight-site system. The isometry center of a TTN is marked with a white cross, and dashed boxes surround tensors that are being contracted. When isometrized, all the tensors apart from the isometry center are unitary and thus a priori known to contract into identities. Therefore, the norm of the isometry center tensor corresponds to the norm of an entire tensor network.

## 3.2   Disentanglers

A disentangler is a two-body unitary gate attached to a pair of sites of the tensor network. The disentanglers were originally introduced in the context of the multiscale entanglement renormalization ansatz (MERA) [14, 18]. The aTTN is, in fact, a subclass of MERA with one incomplete disentangler layer. The restrictions and advice on how to best position the disentanglers in the disentangler layer are explained in Sec. 6.1. We depict disentanglers in the figures with green rectangles.

## 3.3  Matrix product operators

A matrix product operator (MPO) is matrix product state's operator counterpart. A general MPO is depicted in Fig. 5(a) for an example of a four-body system. There are different ways an MPO can be represented and implemented, e.g., dense MPOs, sparse MPOs, etc. [32, 33]. When the operator we want to represent is a sum of $n$-body terms, $\text{MPO} = \sum_\alpha \hat{h}_\alpha$, we can store its MPO as a set of tensor product operator (TPO) terms $\hat{h}_\alpha$ [25]. As described in Sec. 4.1, our aTTN ground state optimization algorithm relies on the TPO representation of the Hamiltonian MPO. An example of the TPO-representation of a four-body MPO consisting of local and 2-body terms is shown in Fig. 5(b).

From now on, the TPO representation is implicit for all the MPOs throughout the cookbook, even if the MPO is for simplicity depicted as a general MPO as in Fig. 5a). Moreover, all the MPO sites, including the ones on the edges, are depicted with two horizontal links, one left and one right, even though the edge link may just be a dummy link of dimension one. Implementation-wise, placing the dummy links on edge MPO sites and keeping the same number of horizontal links for all MPO sites reduces the number of if-cases in the code and moreover, allows to handle the case with periodic boundary conditions. In this section, we define three subtypes of MPOs relevant for the aTTN ground state search algorithm: the Hamiltonian, the effective operators, and the environments.

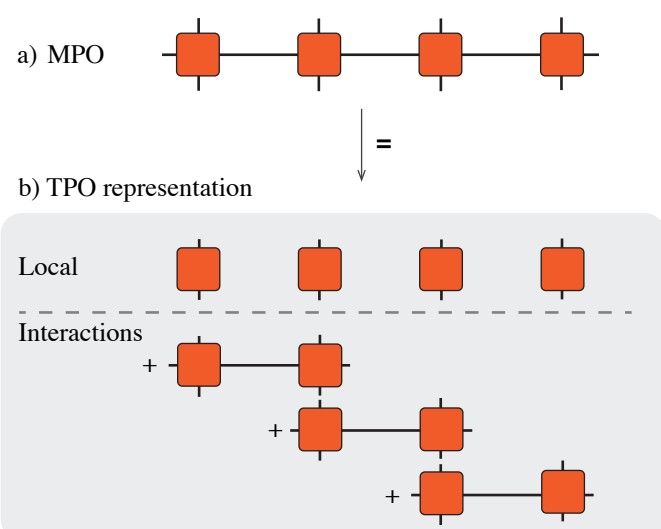

Figure 5: **Matrix product operators (MPOs).** (a) A general four-body MPO. (b) Example of the TPO representation of a four-body MPO, e.g., Hamiltonian, consisting of local and two-body terms. The TPO representation implies that an MPO is stored as a set of individual terms, i.e., TPO terms. All local terms acting on the same site can always be contracted and stored as a single local term.

### 3.3.1  Matrix product operator for the system Hamiltonian

We consider Hamiltonians with the structure $\hat{H} = \sum_\alpha \hat{h}_\alpha$, with $\hat{h}_\alpha$ being n-body terms with any range. Here, the choice of a TPO representation is very handy in terms of the flexibility of writing a Hamiltonian MPO. While the dense or sparse MPO representation can be done by manually constructing MPO tensors for each Hamiltonian based on case-specific rules, storing the Hamiltonian term-by-term allows for constructing it directly from the definition of any model. The TPO representation encodes the MPO exactly, with no truncation on the level

of the Hamiltonian. Tensors of the Hamiltonian-TPOs are depicted in figures as dark orange squares or rectangles, see Fig. 5.

### 3.3.2 Effective operators

Many tensor network algorithms, such as the density matrix renormalization group (DMRG) [7] or the time-dependent variational principle (TDVP) [34, 35], rely on the computation of the effective operators, defined as the effective MPO terms which act on a single tensor in a tensor network. Given an operator whose expectation value we are computing, the effective operators are obtained by contracting the surrounding tensor network as in Fig. 6 on the example of a TTN. Before the contraction, the isometry center is moved to the tensor whose effective operators we are computing. Each tensor in a TTN has three corresponding effective operators, as illustrated in the last step in Fig. 6. Here, we denote effective operators with yellow rectangles in the figures. Throughout the paper, referring to the effective operators of an aTTN implies the effective operators of a corresponding TTN part.

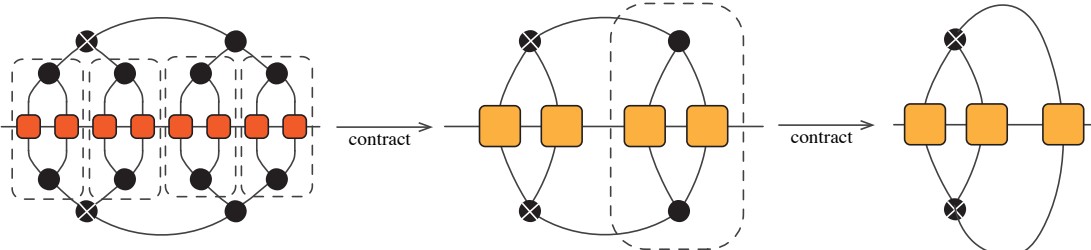

Figure 6: **Obtaining the effective operators of a TTN.** The effective operators (yellow) are the effective MPO terms acting on a single tensor in a tensor network state, arising during the contraction of the MPO operator (red) expectation value. The figure shows the contraction procedure for obtaining the effective operators of the topleft TTN tensor. The analog procedure is performed to obtain the effective operators of any tensor in a TTN, resulting in three effective operators per tensor. Notice that the MPO terms are the effective operators of the lowest-layer tensors. The dashed rectangles denote the tensor contraction.

It is useful to explain how we handle the effective operators using the TPO structure. The contractions as in Fig. 6 are carried out separately for each TPO term, keeping the TPO structure until reaching the last stage in Fig. 6. Therefore, the effective operators around a single tensor consist of a list of TPO terms. All the TPO terms inside the effective operators are then passed as an input to a generalized matrix-vector multiplication function required for the Krylov solver in variational algorithms. Inside the matrix-vector multiplication function, each of the input TPO terms is consecutively contracted to the tensor. The scheme benefits computationally from the fact that the bond dimension between the TPO tensors inside the same TPO term is usually equal to one.

### 3.3.3 Environments

We define the environment as the MPO term with one or more uncontracted links pointing to the child tensor, unlike the MPO term for effective operators, whose uncontracted links point to the parent tensor or to another MPO term. Like the disentanglers, the concept of the environment originates from the MERA algorithms [18]. The name comes from the fact that the environment surrounds a tensor or a group of tensors in the network. An example is shown in Fig. 7, where we depict the environment as a turquoise shape. The environments appear

during the contractions in the aTTN optimization algorithm, alongside effective operators, TTN tensors, and disentanglers.

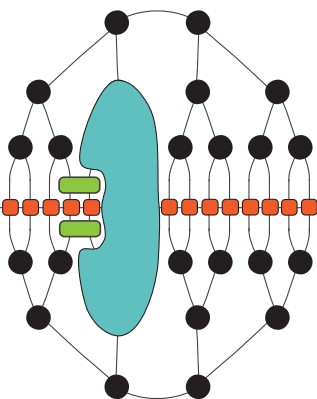

Figure 7: **An example of the environment tensor in the aTTN.** An environment tensor is an MPO term with one or more uncontracted child links. In the figure, the uncontracted child link points towards the disentangler.

## 4   Recipe 1: Ground state search

Suppose we are interested in the ground state of the Hamiltonian $\hat{H}$. We find the ground state energy and ground state by minimizing the energy:

$$E = \langle \psi | \hat{H} | \psi \rangle \, ; \;\; \langle \psi | \psi \rangle = 1, \tag{2}$$

over all the possible states $|\psi\rangle$ in a parameter space spanned by a certain ansatz. In our case, the ansatz is the aTTN $|\psi_{\text{aTTN}}\rangle$, which contains two unknowns to optimize: the disentanglers $D(u)$ and the TTN $|\psi_{\text{TTN}}\rangle$. The expectation value of the energy $\langle \psi_{\text{aTTN}} | \hat{H} | \psi_{\text{aTTN}} \rangle$ in graphical notation is shown in Fig. 8.

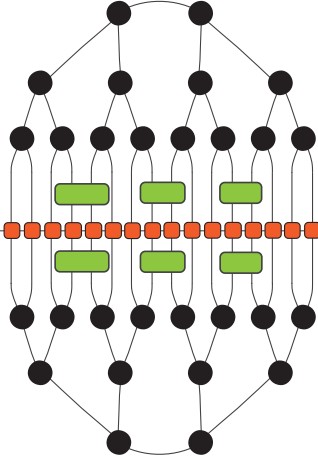

Figure 8: **Expectation value of an MPO with the aTTN ansatz.** In the ground state search, we are minimizing the expectation value of the Hamiltonian MPO $\langle \psi_{\text{aTTN}} | \hat{H} | \psi_{\text{aTTN}} \rangle$.

A standard approach to finding the ground state with tensor networks is the variational DMRG algorithm, or alternatively, the imaginary time evolution [36] via time evolution algorithms [37]. In the variational DMRG, we minimize the energy contribution of each tensor while all other are kept constant. This is a variational procedure, which can be iterated until convergence. The ground state search for the aTTN consists of a DMRG sweep of the TTN, followed by an optimization of the disentanglers. Each aTTN optimization sweep is thus split into three parts:

1. Optimize all disentanglers in $D(u)$;

2. Map the Hamiltonian to an auxiliary one $\hat{H} \longrightarrow \hat{H}' = D(u)\hat{H}D^{\dagger}(u)$;

3. Run a variational DMRG sweep for finding the optimal TTN state $|\psi_{TTN}\rangle$ for the auxiliary Hamiltonian $\hat{H}'$.

The procedure is iterated, updating the disentanglers and finding a new optimal TTN throughout the sweeps. In practice, the sweep can be performed without updating the disentanglers. As we show in Sec. 6.2, we obtain the biggest advantage from the disentangler optimization when running the first sweep without the disentanglers and all the following sweeps, including the disentangler optimization. In Secs. 4.1-4.3, we explain the first and the second step of the algorithm, i.e., a technique for finding the optimal disentanglers and the mapping of the Hamiltonian to an auxiliary one. For a guide to the variational TTN ground state search, see the references [2, 22, 26, 38].

## 4.1 Optimizing the disentangler layer

We search for the optimal set of disentanglers by optimizing each disentangler $u_k \in D(u)$ individually. Suppose that we are optimizing the disentangler $u_k$. The total energy of the state is obtained by summing up the contributions of every TPO term in a Hamiltonian (Fig. 9(a)). These can be split into two parts:

$$E = \sum_p E_k^p(u_k) + c_k. \tag{3}$$

As illustrated in Fig. 9(b), the first part sums the energy contributions of all the TPO terms which act on any of the two sites connected to $u_k$, and thus depend on the disentangler, while the second term contains all remaining TPO terms. The contribution of the latter is independent on $u_k$, and can thus be discarded during the optimization.

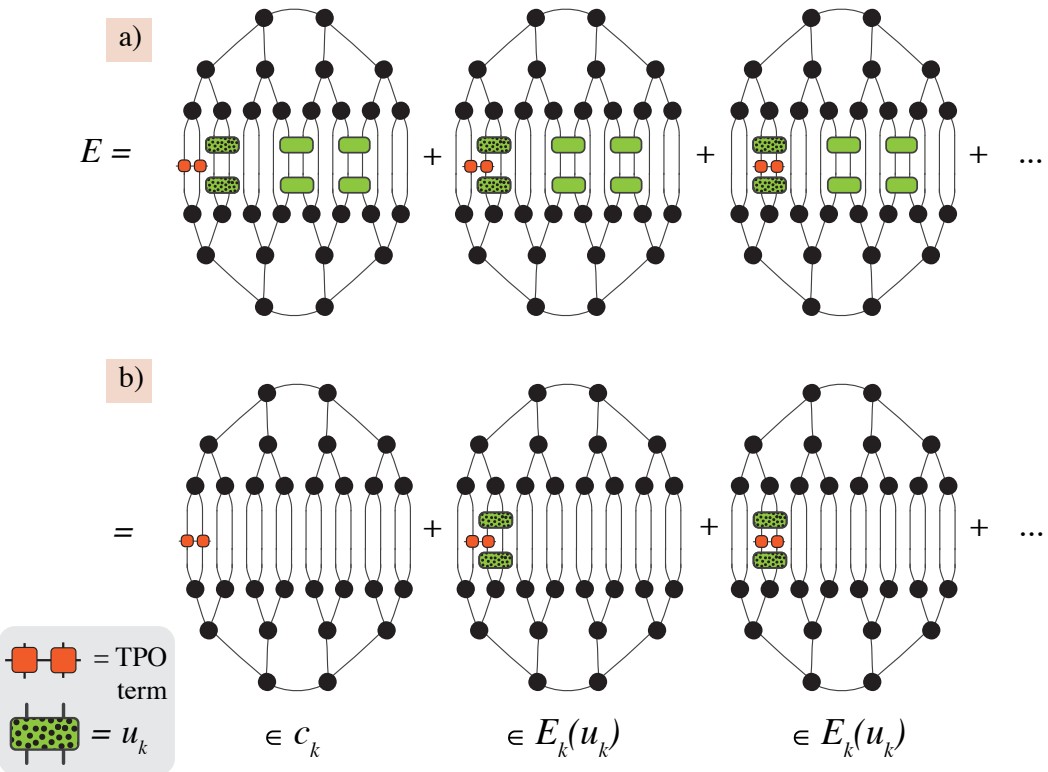

Figure 9: **Computing the aTTN energy with a TPO Hamiltonian.** (a) The total energy of an aTTN state is obtained by summing up the energy contributions from every TPO term in the Hamiltonian. Here, we illustrate the first three terms in the sum on the example of the nearest-neighbour Hamiltonian with two-body interactions. (b) We can split the contributions into two parts: those of the TPO terms which act on the disentangler $u_k$ shaded with dots ($\equiv E_k(u_k)$), and those of the TPO terms which do not act on the disentangler ($\equiv c_k$). Due to their unitarity, the disentanglers cancel out if not attached to the TPO term, whose energy contribution we are computing.

Now, we recast $\sum_p E_k^p(u_k)$ as a cost function in a form suitable for the minimization. The first step is to contract the tensor network around a single disentangler $u_k$ and its Hermitian conjugate $u_k^\dagger$, to obtain a tensor network as in Fig. 10. The surrounding tensors are the left and the right environment of the disentangler. For now, we skip the details on how to obtain the environments and focus on the main steps of the algorithm (the procedure is described in detail in Sec. 4.2). If we choose the positions of the disentanglers such that no TPO term in the Hamiltonian is connected to more than one disentangler, all the remaining disentanglers $u_{i\neq k}$ contract to identities with their Hermitian conjugates, so the environments do not depend on any of them. This imposes an important restriction on the positioning of the disentangler: no Hamiltonian TPO term can be connected to more than one disentangler. For a discussion of all restrictions on disentangler positioning, see Sec. 6.1.

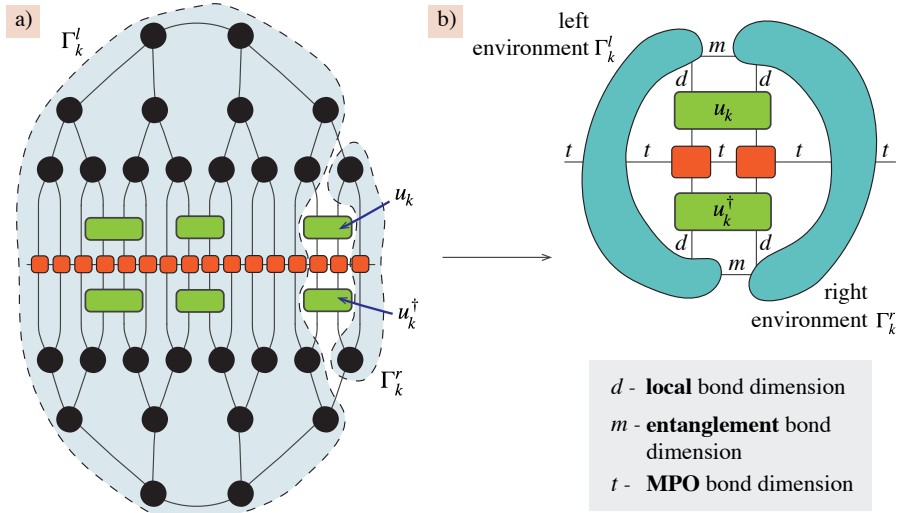

Figure 10: **Main steps in the optimization of the disentangler $u_k$.** (a) First, the tensor network is contracted into two environments, $\Gamma_k^l$ and $\Gamma_k^r$, as indicated by the dashed lines. (b) The tensor network of energy expectation value after the contraction, with indicated bond dimensions.

At the stage as in Fig 10b), there are both $u_k$ and $u_k^\dagger$ present in the tensor network. However, the optimization of $u_k$ and $u_k^\dagger$ simultaneously in this form is a difficult minimization problem. We describe two possible approaches to optimization: the self-consistent optimization like performed in MERA, and an approach based on gradient descent.

### 4.1.1 MERA-like disentangler optimization

This is the approach introduced for the optimization of MERA [18] and used in Refs. [25, 26] for aTTNs: we fix the $u_k^\dagger$ and optimize only $u_k$, then solve the problem self-consistently for a certain number of iterations (Fig. 11). By fixing $u_k^\dagger$, we can contract the tensor network in Fig. 11a) around $u_k$. Contracting the environments with $u_k^\dagger$ and MPOs results in a tensor network as in Fig. 11b), corresponding to an expression of the form:

$$E = \mathrm{Tr}(u_k \Gamma_k). \tag{4}$$

Here, $\Gamma_k$ denotes the global environment obtained by contracting $\Gamma_k^l$, $\Gamma_k^r$, MPO terms, and $u_k^\dagger$. The minimum of Eq. (4) is obtained by choosing $u_k = -VU^\dagger$ such that $U$ and $V$ are the unitary transformations from the singular value decomposition of the global environment $\Gamma_k = U\sigma V^\dagger$ Therefore, Eq. (4) simplifies to:

$$E = Tr(-VU^\dagger U\sigma V^\dagger) = -\sum_j \sigma_j, \tag{5}$$

where $\sigma_j \geq 0$ are singular values of the global environment matrix. Once $u_k$ is optimized, we update the $u_k^\dagger$ and repeat the contraction in Fig. 11 to obtain the new $\Gamma_k$. We repeat the procedure until convergence. Notice that there is no need to recalculate the left and the right environments throughout iterations in the optimization.

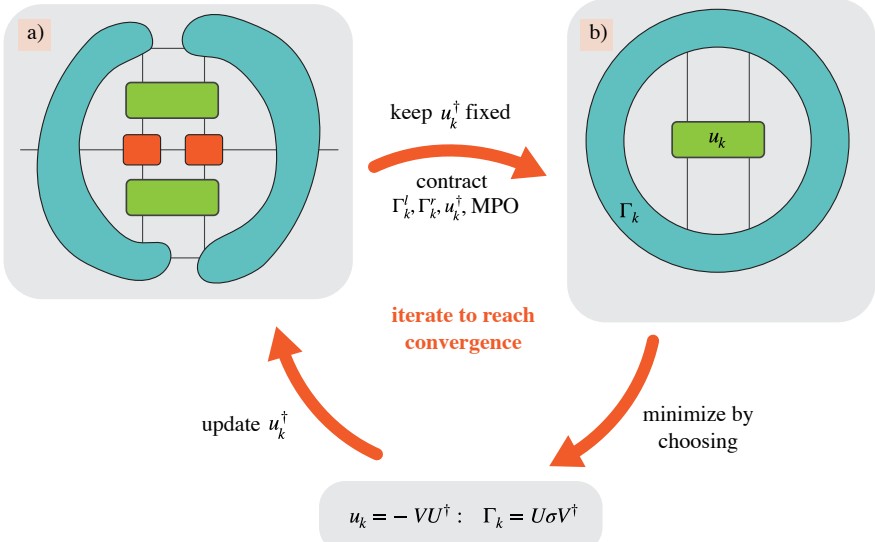

Figure 11: **MERA-like optimization of the disentangler** $u_k$**.** We minimize the energy by fixing $u_k^\dagger$ and optimize only $u_k$, then solve the problem self-consistently. Starting from (a), the environments $\Gamma_k^l$ and $\Gamma_k^r$ and the remaining MPO terms are contracted with complex conjugate disentangler $u_k^\dagger$, yielding a form suitable for minimization of $u_k$ in (b).

### 4.1.2 Disentangler optimization with gradient descent

An alternative approach is optimizing the disentanglers using gradient descent. In this case, both $u_k$ and $u_k^\dagger$ can be optimized simultaneously. We define the cost function as the energy computed by the complete contraction of a tensor network in Fig. 10(b), with elements of $u_k$ as the variational parameters. The problem is nontrivial because the disentangler must satisfy the unitarity constraint, $u_k^\dagger u_k = \mathbb{1}$. We point out a potential approach using the Riemannian optimization with the nonlinear conjugate gradient or quasi-Newton algorithms, shown to perform successfully for optimizing isometric tensor networks [39].

### 4.2 Details on the environment contractions

This section contains the detailed description of the environment contractions in Fig. 10 and Fig. 11. Readers interested only in the high-level description of the algorithm may proceed directly to Sec. 4.3.

### 4.2.1 Obtaining the left and right environment

We identify a unique path through a TTN which connects the two disentangler sites and choose an anchor tensor along it (Fig. 12(a)). We define the left environment as the tensor network connected to the parent and the left child links, including the anchor, and the right environment as the tensor network connected to the right child link of the anchor, excluding the anchor (Fig. 12(b)). Both environments exclude the disentangler and the TPO tensors connected to the disentangler. Here, we stick to the convention that the anchor is the uppermost left tensor on the path, but in general, any tensor on the path can be the anchor.

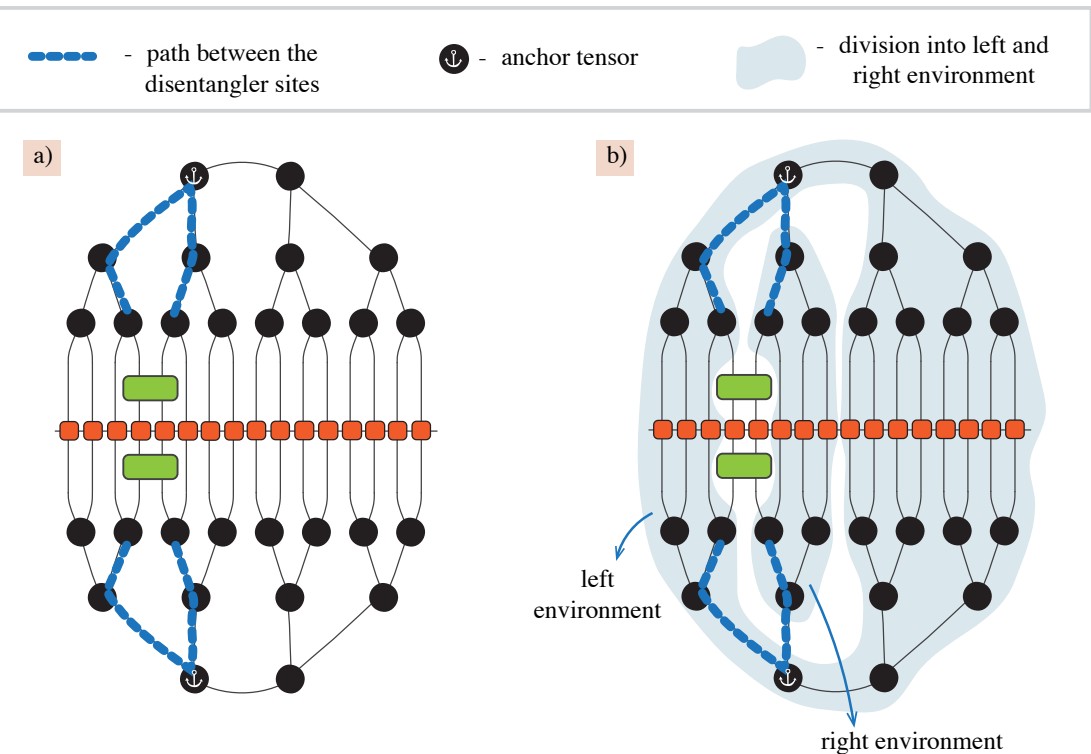

Figure 12: **Convention for division into the left environment $\Gamma_k^l$ and right environment $\Gamma_k^r$.** (a) We first track the path connecting the two disentangler sites through the TTN (blue dashed line) and choose the topmost left tensor along the path as the anchor (denoted with white anchor). (b) The division into the environments is highlighted with light blue. The left environment includes the anchor and the tensor network contracted to its parent and left child links, and the right environment includes the tensor network connected to the right child link of the anchor, excluding the anchor.

Performing the contraction to the environments starts with the initial contraction to obtain the pre-environments, followed by the iterative contraction scheme. First, we move the isometry center to the anchor tensor. Both the left and the right pre-environments are obtained by performing the steps as in Fig. 13.

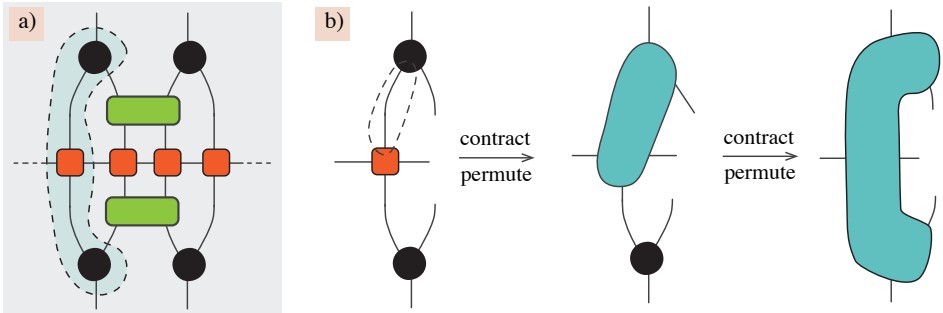

Figure 13: **Obtaining the pre-environments.** (a) To prepare the network in a shape suitable for the iterative contraction scheme, we perform the initial contraction of the tensors surrounded with dashed lines. The disentangler sites are depicted to be the neighbouring sites for the simplicity of the picture; in general, they can be on distant positions. (b) The detailed steps of contraction in (a). We refer to the final tensor as the pre-environment. The procedure is shown for the tree tensor in the left environment, but analogous steps are performed for the tree tensor in the right environment.

After the pre-contraction, we consider the tensors on the path and build all of their effective operators. Then, we calculate the full left and right environments by contracting tensors along the path one by one with the corresponding effective operator (yellow tensor). One step of the iteration is shown in Fig. 14, and iterations along the path are shown in Fig. 15. Recall that we only have to do this contraction along the path, as the rest of the network remains unitary and thus cancels out.

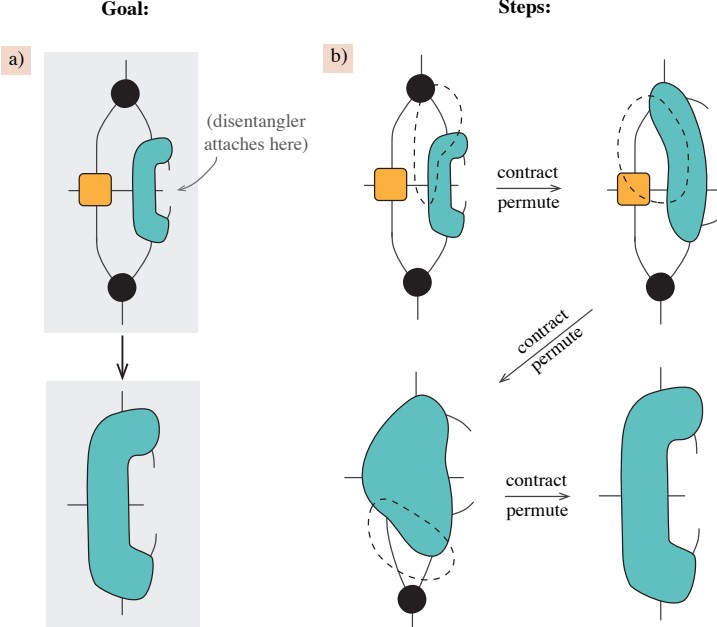

Figure 14: **One step of the iterative environment contraction.** (a) The goal is to contract the (pre-)environment tensor with the connected tree tensors and the effective operator to obtain the new environment of the same link structure. As an iterative procedure, the new environment has the same shape as the input, but has absorbed one more layer via contractions. (b) The detailed steps of the contraction in (a).

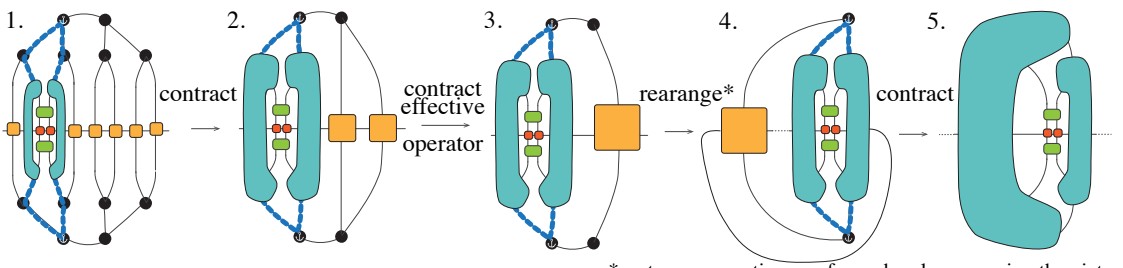

*no tensor operations performed, only rearranging the picture

Figure 15: **Overview of the iterative environment contraction.** Contraction steps
$1 \rightarrow 2$ and $3 \rightarrow 4$ imply a contraction as in Fig. 14 on both left and right environment.
We start by contracting the lowest tensors on the path (blue dashed lines) and move
towards the anchor. By convention, the anchor is included in the left environment.
In step $2 \rightarrow 3$, we build an effective operator and shift it from the right side of the
network to the left side for clarity.

### 4.2.2 Contracting to global environment

If we choose the MERA approach to the disentangler optimization, i.e.,fixing $u_k^{\dagger}$ and optimizing
$u_k$ (Sec. 4.1.1), we have to contract the obtained left and right environment with the rest of
the network (Fig. 11a)-b)) to obtain the matrix $\Gamma_k$. One can think of different ways to carry
out those contractions. In Fig. 16, we show the approach used in the Quantum TEA library,
which minimizes the required memory. We start by contracting the environments, then the
Hermitian conjugate disentangler, and finally the MPO. The biggest tensor constructed in the
process it the full environment, with a maximum of $d^4 t^4$ elements.

Note that optimizing the disentanglers with the gradient descent approach does not re-
quire this step. Moreover, an important remark regarding the MPO bond dimension $t$ is the
following. As mentioned, the MPO Hamiltonian can be composed of an arbitrary combination
of operators acting on arbitrary sites. Hence, the bond dimension $t$ cannot be thought of as
an exact dimension of the depicted bonds, but is rather as an approximate measure allowing
us to get a grasp of the computational complexity. In most cases, $t$ is equal to one.

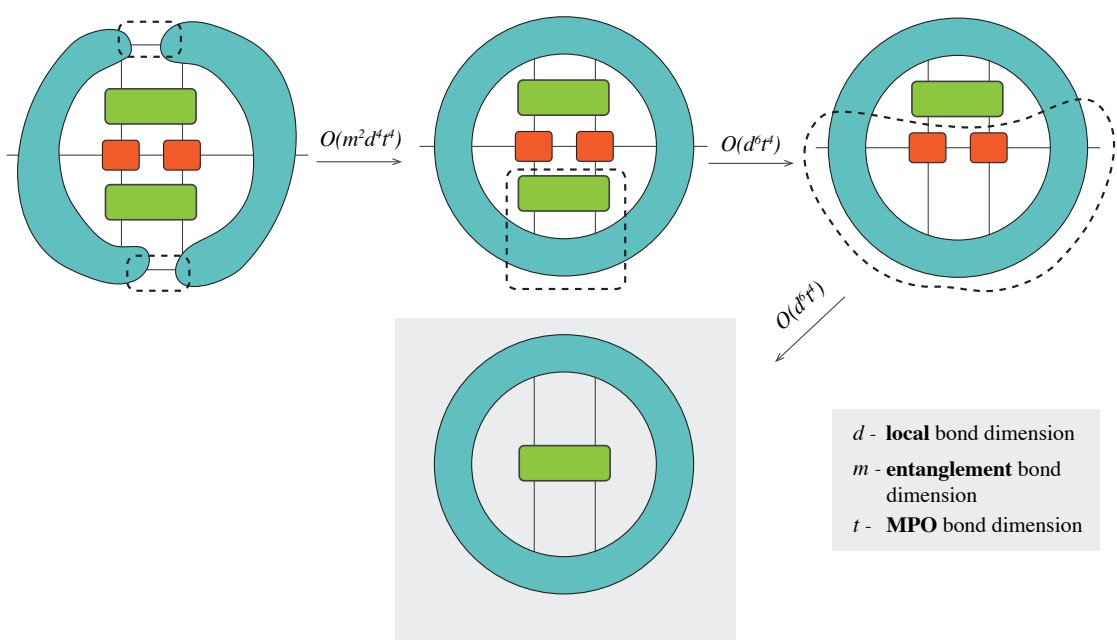

Figure 16: **The contraction to the global environment $\Gamma_k$ with indicated computational complexities of the operations.** In the MERA-like disentangler optimization, we fix $u_k^\dagger$ and contract the entire network around $u_k$. By contracting to the final tensor network, we bring the optimization problem to a form $\text{Tr}(u_k \Gamma_k)$, whose solution for $u_k$ is known.

## 4.3 Updating the Hamiltonian matrix product operator

After the optimization of the disentangler layer, we contract it to the Hamiltonian. This object is then used as the Hamiltonian for the following TTN optimization. Here, we describe how to implement the contraction of the original Hamiltonian $\hat{H}$ to a new, auxiliary one $\hat{H}' = D\hat{H}D^\dagger$. The procedure is general and can be applied to contracting the disentangler layer to any TPO:

---

**function** CONTRACT DISENTANGLER LAYER(TPO, disentangler layer)
  **for** TPO term **in** TPO **do**
    **for** disentangler **in** disentangler layer **do**
      **if** TPO term sites and disentangler sites intersect **then**
        **if** every disentangler site is in TPO term sites **then**
          contract as in Fig 17(a)
        **else**
          contract as in Fig 17(b)
        update the TPO term with the contracted version

---

Before the contraction, we decompose the disentangler into a two-body term using the QR decomposition. If the initial TPO term was an $n$-body term, after the contraction it either remains a $n$-body term (scenario Fig. 17(a)), or becomes a $(n+1)$-body term (scenario Fig. 17(b)). The details of the contraction for a general case are explained in the following subsection. Note that contracting the disentangler with a 2-body TPO term increases the bond dimension of the TPO to at most $d^2$, as indicated in Fig. 17. For $n$-body TPO terms with $n > 2$, contracting the disentanglers in general increases the bond dimension by a factor of $d^4$.

The increased span and bond dimension of the TPO terms are the reasons why the aTTN algorithms are more costly with respect to TTN algorithms. Note that this is a constant prefac-

tor. The scaling of the computational complexity with the bond dimension $m$ does not change when going from a TTN to an aTTN.

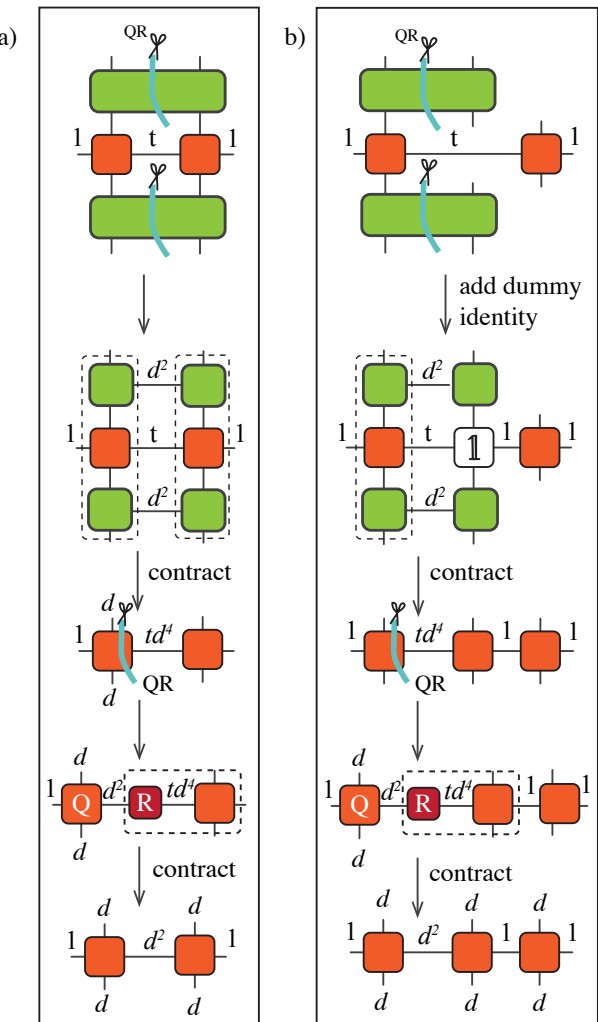

Figure 17: **The contraction of the disentanglers with a TPO term for two possible cases of the TPO term and the relative position of the disentangler.** Depicted in the example of a two-body TPO term (as in standard nearest-neighbour Hamiltonians) with bond dimensions $t$. (a) The disentangler lies fully on the TPO term. The disentanglers are decomposed into two tensors, each of which is contracted with the corresponding leg of the TPO. (b) When one disentangler site does not lie on the TPO term, we expand it with an identity tensor placed on the extra site. After the QR decomposition, the disentangler tensors are contracted with the corresponding legs of the TPO. In both (a) and (b), the TPO bond dimension grows to $t \cdot d^4$ immediately after the disentangler contraction. However, if the initial TPO term is a two-body, as in this example, the bond dimension can always be reduced to $d^2$ using subsequent QR decompositions.

### 4.3.1 Contracting the disentangler with a tensor product operator term

The two sites on which the disentangler acts are not restricted to the nearest neighbour sites in a TTN, as the example in Fig. 17 shows. Therefore, we need a systematic way to handle the extra horizontal link connecting the disentangler sites, and propagate it through the new TPO

term. After inserting the required identity tensors (see Fig. 17(b)), the procedure that handles the general contraction is:

1. Contract the left disentangler site to the corresponding TPO term site (Fig. 18(a)).

2. Propagate the horizontal link through the TPO term sites towards the left disentangler site position, using a series of QR decompositions (Fig. 18(b-e)).

3. Contract the right disentangler site (Fig. 18(f)).

4. Repeat steps (1-3) for the disentangler's conjugate.

Steps 1-3 are shown in Fig. 18 in the example of a four-site TPO term. If using the truncated SVD decomposition instead of the QR-decomposition, we must ensure that the TPO term's isometry center is always at the tensor we are decomposing via SVD.

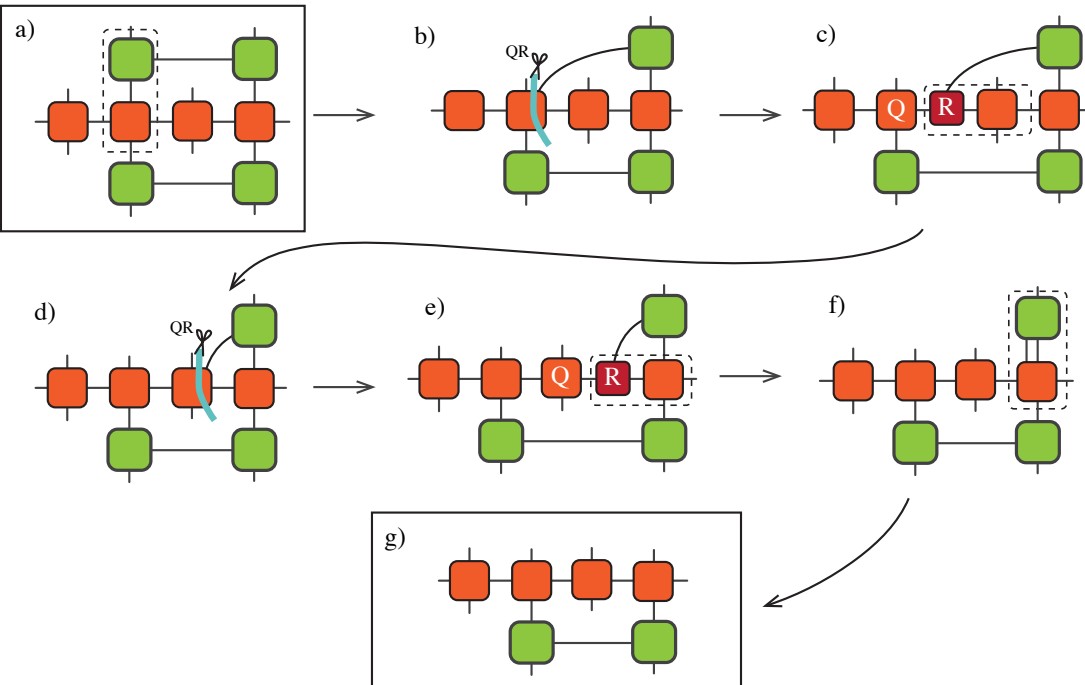

Figure 18: **Steps for contracting the disentangler to a TPO term and propagating the link**. Depicted in an example of a four-body TPO term. (a) Contracting the left disentangler site. (b-e) Propagation of the horizontal links with the QR decompositions. (f) Contracting the right disentangler site. (g) The resulting TPO term after the contraction with the disentangler. The same procedure is applied to the disentangler's conjugate.

## 4.4 Computational complexity overview

We consider one sweep of the aTTN ground state search. The analysis of the computational complexity is split into two parts: the optimization of the disentangler layer and the variational ground state search with the auxiliary Hamiltonian $\hat{H}' = D(u)\hat{H}D^{\dagger}(u)$. As in the previous chapters, $N$ is the total number of sites, $m$ is the maximal bond dimension of the TTN, $d$ is the local Hilbert space dimension, and $t$ is the MPO bond dimension.

### 4.4.1 Complexity of the optimization of the disentangler layer

Each step of the iterative environment contraction (Fig. 14) scales as $\mathcal{O}(m^4 d^2)$. Taking into account that the number of iterations is proportional to the number of layers in a TTN, which grows logarithmically with the system size, the overall complexity is $\mathcal{O}(\log(N) \cdot m^4 d^2)$. The contraction to the global environment $\Gamma_k$ (Fig. 16) is $\mathcal{O}(m^2 d^4) + \mathcal{O}(d^6)$, and the cost of the SVD for $\Gamma_k$ is $\mathcal{O}((d^2)^3) = \mathcal{O}(d^6)$. Let us define $N_D$ as the total number of disentanglers in the disentangler layer. The procedure needs to be repeated for every disentangler and the MERA-like disentangler optimization takes $N_i$ iterations until convergence. Therefore, the total cost of the disentangler layer optimization is $\mathcal{O}(N_D \cdot \log(N) \cdot m^4 d^2) + \mathcal{O}(N_D \cdot N_i \cdot m^2 d^4) + \mathcal{O}(N_D \cdot N_i \cdot d^6)$. Moreover, assuming that $d \ll m$ and $N_i d^2 \ll m^2$, the expression can be reduced to $\mathcal{O}(N_D \cdot \log(N) \cdot m^4 d^2)$. In practice, this part is not the bottleneck of the simulation, as it has a much smaller constant prefactor.

### 4.4.2 Complexity of the variational ground state search with the auxiliary Hamiltonian

A full sweep in the variational ground state search consists of computing the effective operators for each tensor, and solving the local eigenproblem with the Lanczos algorithm. The total computational cost reflects the cost of the local matrix-vector multiplication performed at each iteration of the Lanczos algorithm. This corresponds to the contraction of three effective operators with an order-three tensor as in Fig. 19. For a dense MPO of bond dimension $t$, this cost is $\mathcal{O}(m^4 t^3)$.

In the TPO picture, each of the steps in Fig. 19 needs to be performed sequentially for each TPO term. The complexity thus depends on the individual TPO bond dimension $t$ and the number of TPO terms in a single effective operator. Typically, the number of TPO terms in an effective operator scales linearly with the length of the lattice side $L$ in an $L \times L$ system. While in the TTN algorithm each TPO term has a bond dimension $t = 1$, this might not be the case for the aTTN. The original Hamiltonian is contracted with the disentangler layer, which adds a factor of $d^2$ to the bond dimension for two-body Hamiltonian terms, or $d^4$ for $n$-body Hamiltonian terms, to each TPO link over which the disentangler is placed. Therefore, the difference in computational cost between the TTN and the aTTN algorithms scales only with the local dimension $d$. The exact factor depends on the model and the positions of the disentanglers in the lattice.

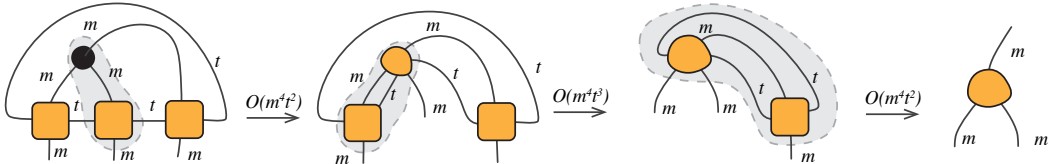

Figure 19: **Computational cost of applying the effective operators to a tree tensor in the Lanczos algorithm.** The figure shows steps of contracting the three effective operators to a tree tensor with indicated bond dimensions. The complete contraction is carried out with the computational cost of $\mathcal{O}(m^4 t^3)$.

## 5 Recipe 2: Measurement of observables

Performing the measurement of local observables and $n$-body correlators of an aTTN draws the logic from the TTN measurement to a large extent. For this reason, we discuss both the logic behind the TTN case and highlight the differences to the aTTN measurements.

### 5.1 Measuring local versus non-local observables with tree tensor networks

Suppose that we want to measure an observable that is a product of local operators on an arbitrary number of sites, in arbitrary positions. In general, it is possible to construct a flexible procedure for measuring the expectation value of any $n$-body operator. For the local observables specifically, i.e., an expectation value of a single-site operator, we construct an efficient computation with the reduced density matrices, as depicted in Fig. 20(a). When a TTN is isometrized towards the lowest-layer tensor corresponding to the site $i$, the isometry center tensor contracted with its complex conjugate represents the reduced density matrix $\rho_i$ of site $i$ (see Fig. 20). The value of the local operator on this site is then straightforwardly computed by contracting the reduced density matrix as in Fig. 20(a). It is enough to compute the reduced density matrices at a certain site once and keep them stored for measuring as many local observables as required.

Apart from the specific case of local observable measurement, all the other $n$-body observables (with $n > 1$) are measured by contracting the tensor network representing the expectation value, as shown in Fig. 20(b). Even the measurement of $n$-body observables benefits from the isometrization, as it is sufficient to contract only the sub-tree containing all of the sites over which the $n$-body observable spans.

### 5.2 Measurement of an observable with the augmented tree tensor network

In the case of an aTTN, the computation of an expectation value needs to include the disentanglers:

$$\langle \hat{o} \rangle_{\mathrm{aTTN}} = \langle \psi_{\mathrm{aTTN}} | \, \hat{o} \, | \psi_{\mathrm{aTTN}} \rangle = \langle \psi_{\mathrm{TTN}} | \, D(u) \, \hat{o} \, D^{\dagger}(u) | \psi_{\mathrm{TTN}} \rangle \equiv \langle \psi_{\mathrm{TTN}} | \, \hat{o}' \, | \psi_{\mathrm{TTN}} \rangle , \qquad (6)$$

This is a specific case of $D(u)$ and $D^{\dagger}(u)$ being connected to an MPO, which we discussed in Sec. 4.3. We perform the measurement by contracting the TPO term $\hat{o}$ with the disentangler layer, $\hat{o}' = D\hat{o}D^{\dagger}$, just as we did with the Hamiltonian terms in the ground state search. Then, we measure $\hat{o}'$ on the TTN part of the aTTN. In Sec. 5.3, we discuss in detail the specific example of a two-body correlation matrix, which is a commonly measured observable.

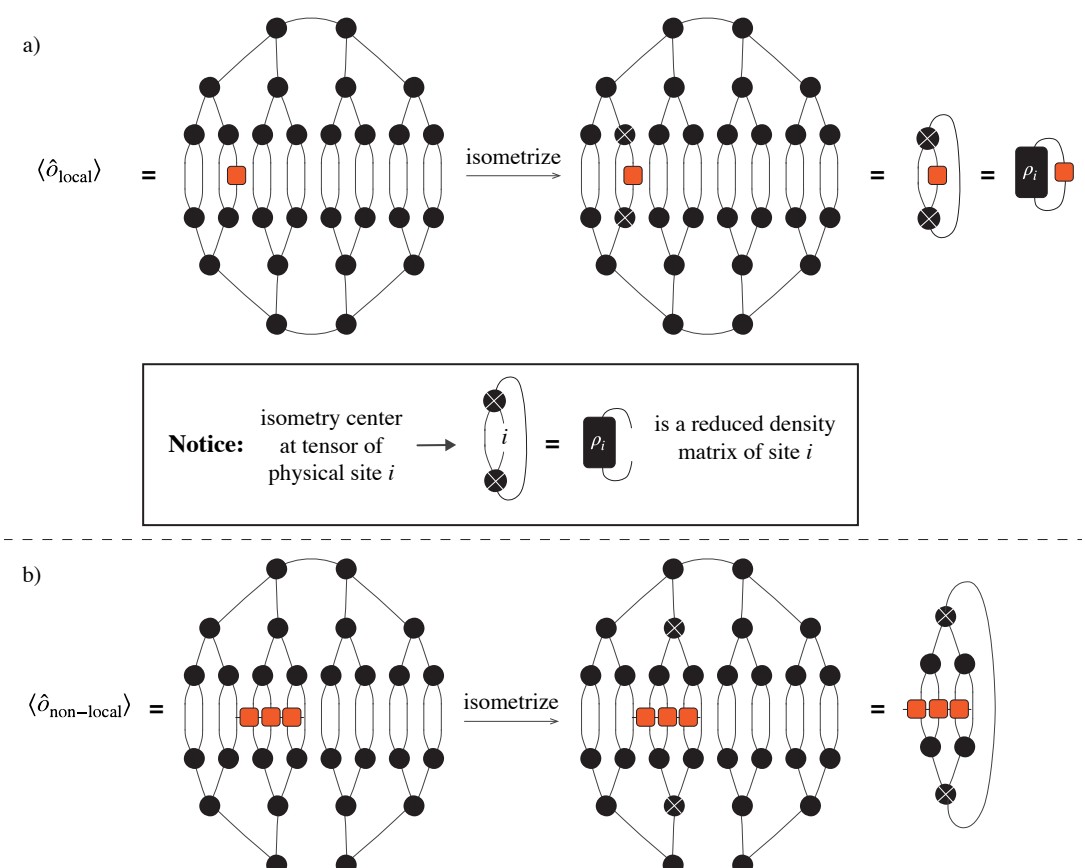

Figure 20: **Measuring the expectation value of local and non-local operators with a TTN.** (a) Measuring a local operator $\langle \hat{o}_{\text{local}} \rangle$. By shifting the isometry center to the tensor of a physical site $i$ with a local term, computing the expectation value reduces to the contraction of the local operator with the tensors of the isometry center, i.e., with the reduced density matrix $\rho_i$. (b) Measuring a non-local operator depicted on the example of a three-body term $\langle \hat{o}_{\text{non-local}} \rangle$. Computing the expectation value can still, to some extent, benefit from isometrization. We need to perform the explicit contraction only for the sub-tree that contains the sites over which the non-local observable spans.

## 5.3   Two-body correlation measurement

Suppose we want to measure the two-body correlation composed of observables $\hat{o}^a$ and $\hat{o}^b$. The output of this measurement for a system consisting of $N$ sites is the $N \times N$ correlation matrix $C$:

$$C = \begin{bmatrix} \langle \hat{o}_1^a \hat{o}_1^b \rangle & \langle \hat{o}_1^a \hat{o}_2^b \rangle & \cdots \\ \\ \langle \hat{o}_1^a \hat{o}_2^b \rangle & \cdots & \cdots \\ \\ \cdots & \cdots & \langle \hat{o}_N^a \hat{o}_N^b \rangle \end{bmatrix}. \tag{7}$$

The subscripts denote the site on which the operator is acting. We treat each of the elements of the matrix $C$ as a TPO term whose expectation value we are computing. In the correlation

matrix $C$, we distinguish between two types of terms: the diagonal terms $\langle \hat{o}_i^a \hat{o}_i^b \rangle$, and off-diagonal terms $\langle \hat{o}_i^a \hat{o}_j^b \rangle$, $i \neq j$. The diagonal terms are, by definition, local. After the contraction with the disentangler layer, they can either stay local or grow to a two-body term (Fig. 21(a)). The off-diagonal terms are two-body terms. After the contraction with the disentangler layer, they can either stay two-body terms, or grow to three- or four-body terms (Fig. 21(b)). The measurement procedure is summarized in Fig. 22 for both the TTN and the aTTN.

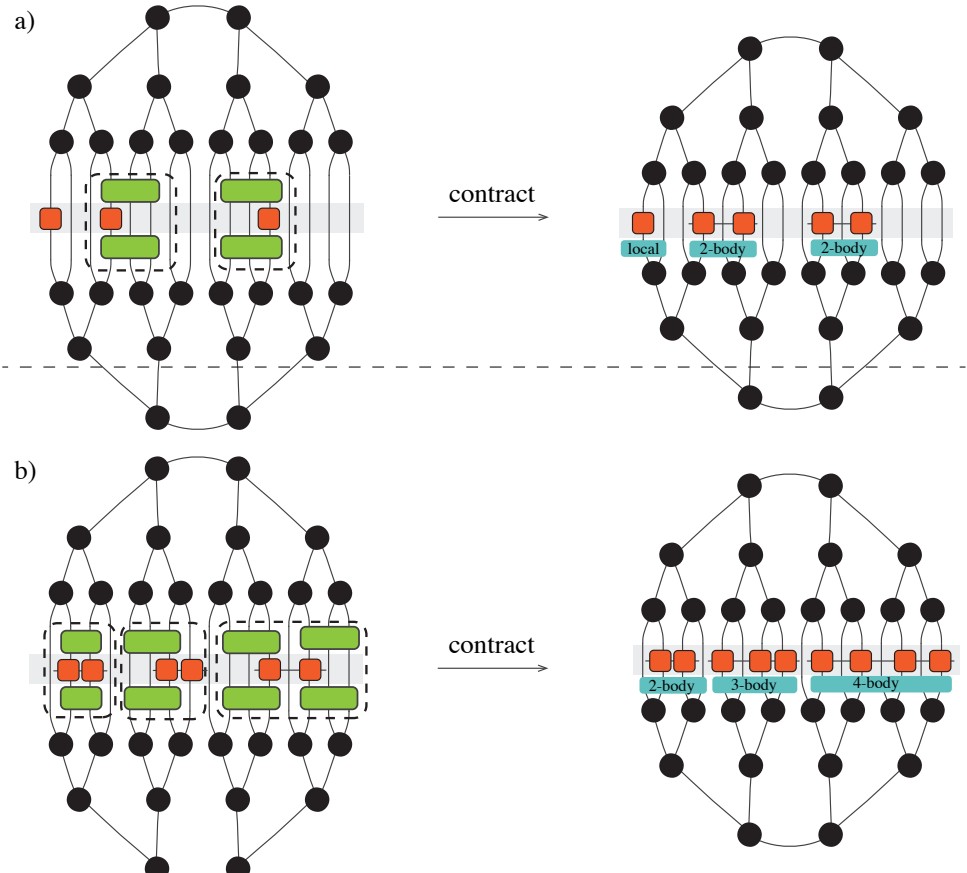

Figure 21: **Different outcomes of contracting the disentanglers with local and two-body terms.** (a) Contracting the disentangler layer with the local terms. The outcome of the contraction is either a local or a two-body term. (b) Contraction of the disentangler layer with the two-body terms. The outcome of the contraction is either a two-body, a three-body, or a four-body term, depending on the number of disentanglers they connect to.

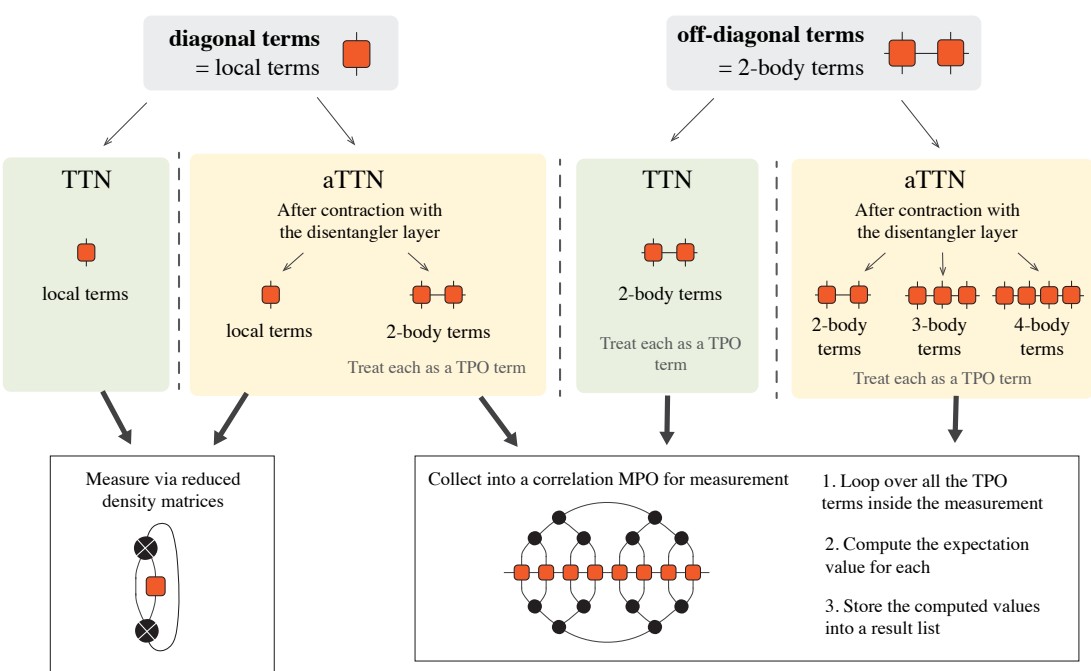

Figure 22: **Overview of the measurement of the two-body correlation matrix for a TTN versus an aTTN**. Diagonal terms refer to terms $\langle \hat{o}_i^a \hat{o}_i^b \rangle$, and off-diagonal terms refer to terms $\langle \hat{o}_i^a \hat{o}_j^b \rangle$, $i \neq j$ in the correlation matrix in Eq. (7).

## 6 Tips for serving

The sections up to now cover a theoretical guide of the algorithms for the optimization and the measurements. Next, we give some advice and insights into the algorithm's performance for different parameters. Namely, we discuss the optimal strategy for positioning the disentanglers and the restrictions in placing them. Then, we compare the performance of the ground state search for MPS, TTN, and aTTN with different bond dimensions, for different system sizes and at different points in a phase diagram. We proceed with the analysis of the results for different numbers of sweeps with disentangler optimization. All results presented further are obtained using the Quantum TEA software package. Simulations are performed on CINECA's HPC LEONARDO, equipped with NVIDIA Ampere A100 GPU and Intel Xeon Platinum 8358 CPU processors. We performed 30 DMRG sweeps for each ground state search. Prior to optimization, all the disentanglers are initialized as identities. To simulate two-dimensional systems, we map the model onto an equivalent one-dimensional one using the Hilbert curve mapping pattern [40].

The models we use for benchmarking are the quantum Ising model on a square lattice and the Heisenberg model on a triangular lattice, both with open boundary conditions. The Hamiltonian of the quantum Ising model is

$$\hat{H} = -J \left( \sum_{<ij>} \sigma_x^i \sigma_x^j - h \sum_i \sigma_z^i \right), \tag{8}$$

where $\sigma_{\{x,y,z\}}^i$ are the Pauli operators acting on site $i$, $< ij >$ denotes the nearest-neighbour sites, $h$ is the external field, and we take $J$ as the energy unit. For a two-dimensional square lattice, the model undergoes a quantum phase transition with the critical point at $h_c \approx 3.044$

in the thermodynamic limit [41].

The triangular lattice Heisenberg model is an example of a frustrated model, described with the Hamiltonian

$$\hat{H} = J \sum_{<ij>} \left( \sigma_x^i \sigma_x^j + \sigma_y^i \sigma_y^j + \sigma_z^i \sigma_z^j \right), \tag{9}$$

where $< ij >$ denotes the nearest-neighbour sites and $J$ is the energy unit. Again, we set $J = 1$ as the energy unit.

## 6.1 Positioning the disentanglers

The number of disentanglers and their positions influence the accuracy and the runtime of the aTTN optimization. We first explain restrictions on placing the disentanglers, and then proceed to describe the strategy for optimal positioning.

### 6.1.1 Restrictions on positioning the disentanglers

We pose two general restrictions on disentangler positioning:

1. **The same TPO (Hamiltonian interaction) term cannot have more than one disentangler attached to it** (Fig. 23(a)). As explained in Sec. 4.1, this restriction is inherent to the optimization algorithm.

2. **The disentangler should not support untruncated links**, i.e., links whose maximal possible bond dimension is smaller than the maximal bond dimension $m$ set in the simulation (Fig. 23(b)). This restriction is not prohibited by the algorithm construction, but rather prevents us from placing the disentanglers on positions that cannot give a gain in energy. Note that this implies that a disentangler cannot be placed on the sites that share the same lowest-layer tensor.

(a)   **Restriction 1**: No more than one disentangler attached to a single TPO term

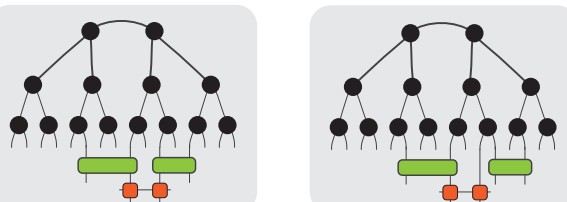

(b)      **Restriction 2**: No disentanglers supporting only untruncated links

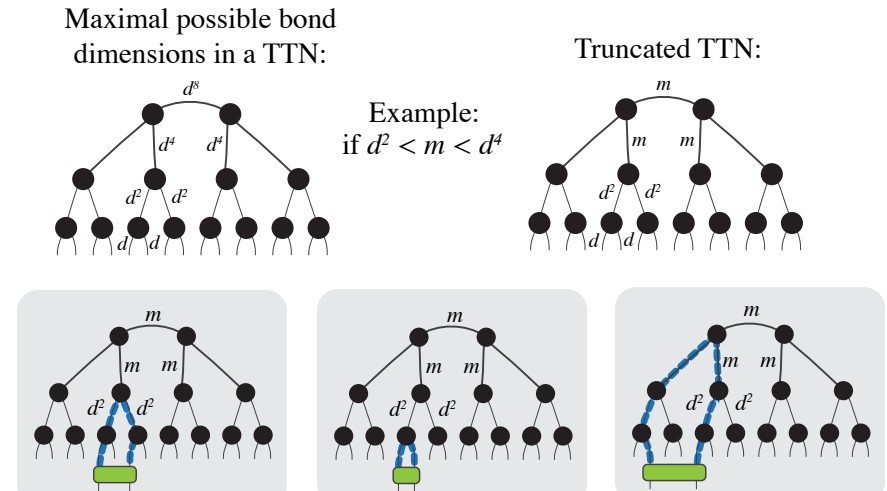

Figure 23: **Restrictions on the disentangler positioning.** (a) The same TPO term cannot have more than one disentangler attached to it: examples of forbidden and allowed disentangler configurations. The condition needs to be satisfied for every TPO term in a Hamiltonian. (b) We avoid placing disentanglers on positions that cannot give a gain in energy because the connecting TTN links already have the maximal possible bond dimension. For a 16-site TTN, we mark the maximal possible TTN bond dimensions (left) and bond dimensions after truncation (right), for example, when $d^2 < m < d^4$, $m$ being the maximal bond dimension set in a simulation. Below, we show examples of forbidden and allowed configurations.

### 6.1.2   Automatic disentangler positioning

An analysis carried out in Ref. [25] has shown that for an $L \times L$ system with a translationally symmetrical Hamiltonian, a successful strategy for positioning the disentanglers can be to place as many disentanglers as possible to support the highest-layer link. These are the links that usually support the largest amount of entanglement. Our strategy is to loop over all of the possible disentangler positions, starting from the ones supporting the highest-layer link, and accept the positions which lie fully on at least one Hamiltonian interaction term (see Fig. 24), and do not violate any of the restrictions from Sec. 6.1.1. An example of the resulting disentangler layer is shown in Fig. 25 for a $32 \times 32$ lattice and the quantum Ising model from Eq. (8).

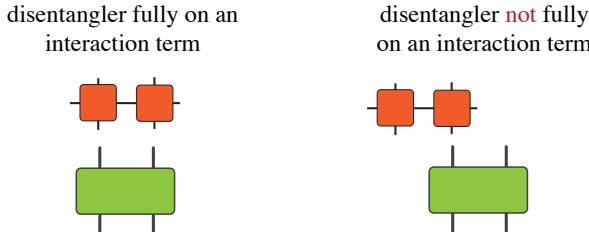

disentangler fully on an
interaction term

disentangler not fully
on an interaction term

Figure 24: **An example of disentangler positions with respect to a single inter-action term**. In the left example, the disentangler lies fully on the interaction term. In the right example, it does not. Our results show that a successful strategy for positioning the disentanglers is to place only those that lie fully on at least one of the interaction terms in the Hamiltonian.

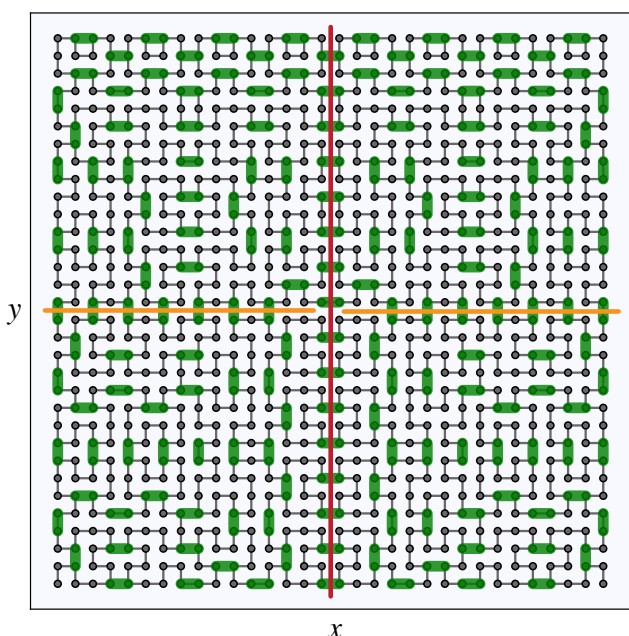

Figure 25: **Positions of the disentanglers for** $32 \times 32$ **lattice quantum Ising model, placed according to the strategy described in text**. The lattice sites and the underlying Hilbert curve mapping are represented with gray dots and lines, and the disentanglers are marked with green. The red and orange lines cut the highest and second-highest layer links of a TTN, respectively.

Keeping only the terms that lie fully on a single interaction term reduces the simulation runtime and also gives a significantly lower energy, as demonstrated in Fig. 26. We show how the resulting energy density $\varepsilon = E/L^2$ (Fig. 26 (a)) and runtime (Fig. 26 (b)) depend on the number of disentanglers computed for the $32 \times 32$ quantum Ising model close to the critical point. Outside of the dark violet region, we place the disentanglers according to the described strategy, allowing only the positions that lie fully on at least one interaction term. The dark violet region shows the regime in which we keep placing the remaining disentanglers, starting once again from the highest-layer bonds, but including also those which do not fully lie on any interaction term. As soon as the latter regime is entered, we clearly see both a jump in the optimized energy value and in the computational time.

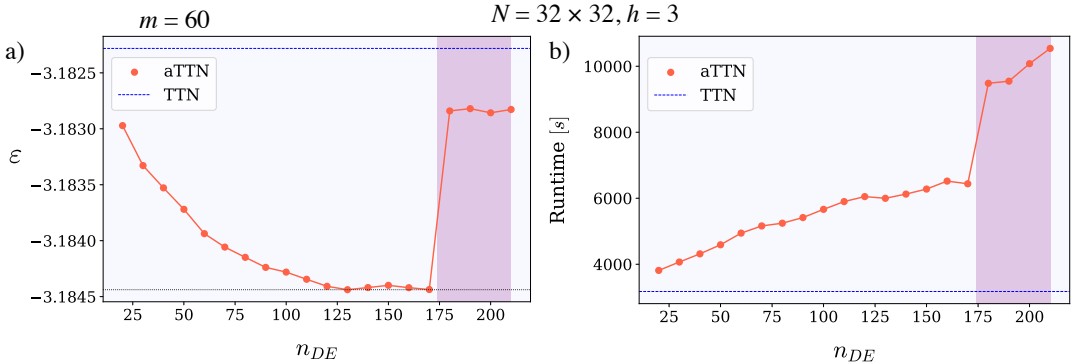

Figure 26: **Ground state search for the quantum Ising model on a** $32 \times 32$ **square lattice, with a varying number of disentanglers in the disentangler layer,** $n_{DE}$**.** We use bond dimension $m = 60$ and choose the external field value $h = 3$, close to the critical point. (a) Ground state energy density $\varepsilon$ as a function of $n_{DE}$. The black dotted line marks the minimum obtained energy density. (b) The corresponding runtime as a function of $n_{DE}$. The simulations are run on a GPU. In both plots, the blue dashed line corresponds to the TTN data obtained with the same bond dimension. A given number of the disentanglers $n_{DE}$ is placed by prioritizing positions according to the strategy described in the text. Outside the dark violet region, we place only the disentanglers that lie fully on at least one interaction term of the Hamiltonian. Inside the dark violet region, we include additional disentanglers that do not lie fully on any interaction term. We observe a sudden decrease in accuracy and an increase in runtime as soon as the dark violet region is entered.

## 6.2 Comparison to matrix product states and tree tensor networks

We benchmark the accuracy of the ground state search for two-dimensional models for different ansätze. We start with the quantum Ising model defined in Eq. (8) on a $32 \times 32$ square lattice and then proceed to the Heisenberg model on a triangular lattice defined in Eq. (9).

### 6.2.1 Quantum Ising model on a square lattice

Fig. 27 (a) shows the difference between the energy densities obtained with the TTN and the aTTN for different values of the transverse field $h$. The bond dimension is fixed to $m = 100$. We see that the region in which the aTTN gives the largest advantage is close to the critical point $h_c \sim 3$. This is where the state's entanglement is the largest, and the model is usually the most demanding to simulate. For $h \approx 1$, in the bulk of the phase, the area law is obeyed. The obtained results are already converged with the TTN within $10^{-7}$ in energy density (Fig. 27 (b)). The difference between the TTN and aTTN energy densities is therefore small.

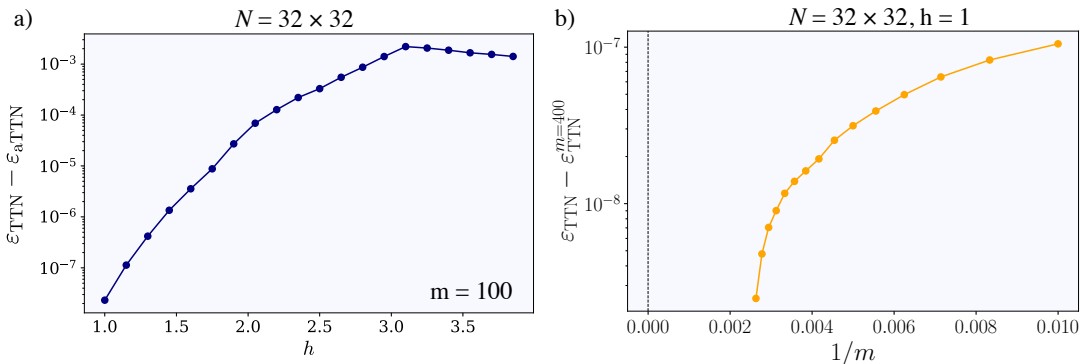

Figure 27: **The comparison between the TTN and the aTTN for the quantum Ising model on a $32 \times 32$ lattice for varying external field $h$ and a fixed bond dimension**. (a) The difference between the TTN and aTTN ground state energy densities, $\varepsilon_{\text{TTN}} - \varepsilon_{\text{aTTN}}$, for bond dimension $m = 100$. (b) The convergence of the TTN energy density $\varepsilon_{\text{TTN}}$ at $h = 1$. The plot shows the difference between the TTN energy density at bond dimension $m$ and the TTN energy density at $m = 400$, plotted as a function of inverse bond dimension $1/m$. The results imply that far away from the criticality, values in (a) are converged within approximately $10^{-7}$.

While in Fig. 27 we have shown that the aTTN always gives an advantage in accuracy for a fixed bond dimension, the required memory and runtime are larger compared to TTN and MPS. Since the amount of computational resources is limited in practice, the following benchmark compares the performance of the aTTN with the TTN and the MPS across different bond dimensions. We evaluate the accuracy, the runtime, and the peak memory allocated on a GPU. The latter is measured with the PyTorch function `torch.cuda.max_memory_allocated()`. The largest bond dimension shown corresponds to the largest possible with the assigned memory resources. Each simulation is assigned 4 CPU cores with a total of 450 GB RAM and a GPU of 64 GB. A GPU offers a significant speedup [42], but poses a limitation in terms of available memory. To balance between those two, we utilize the mixed device mode available in Quantum TEA: during the DMRG sweep, only the isometry center tensor and the corresponding effective operators are stored on a GPU, thus all the operations involving these tensors are carried out on a GPU. This includes solving the local eigenproblem with the Lanczos algorithm. All remaining operations, including the disentangler optimization for the aTTN, is carried out on a CPU host. This way, we perform only the computationally most demanding steps on a GPU, while the rest is kept on a CPU host.

Fig. 28 shows the performance of the different ansätze on the quantum Ising model in a $L \times L$ square lattice for $L = 16, 32$ at $h = 3$, where the biggest advantage of aTTNs is expected according to Fig. 27(a). The presented runtimes for the aTTN ground state search include the disentangler optimization. Since the longest simulation runtime is $\sim 12$ hours, the limiting factor preventing us from reaching higher bond dimensions with the aTTN and the TTN is primarily the limited memory on the GPU. MPS simulations are, on the other hand, primarily limited by the memory requirement on the CPU. This difference arises because the size of the MPS tensors is $m^2 d$ in contrast to $m^3$ in the TTN and the aTTN. As a result, a fixed amount of GPU memory can store MPS tensors and their associated Lanczos tensors at significantly higher bond dimensions than those feasible for the TTN and the aTTN. The bottleneck thus shifts from GPU to CPU resources, which are required to store the rest of the tensor network and effective operators.

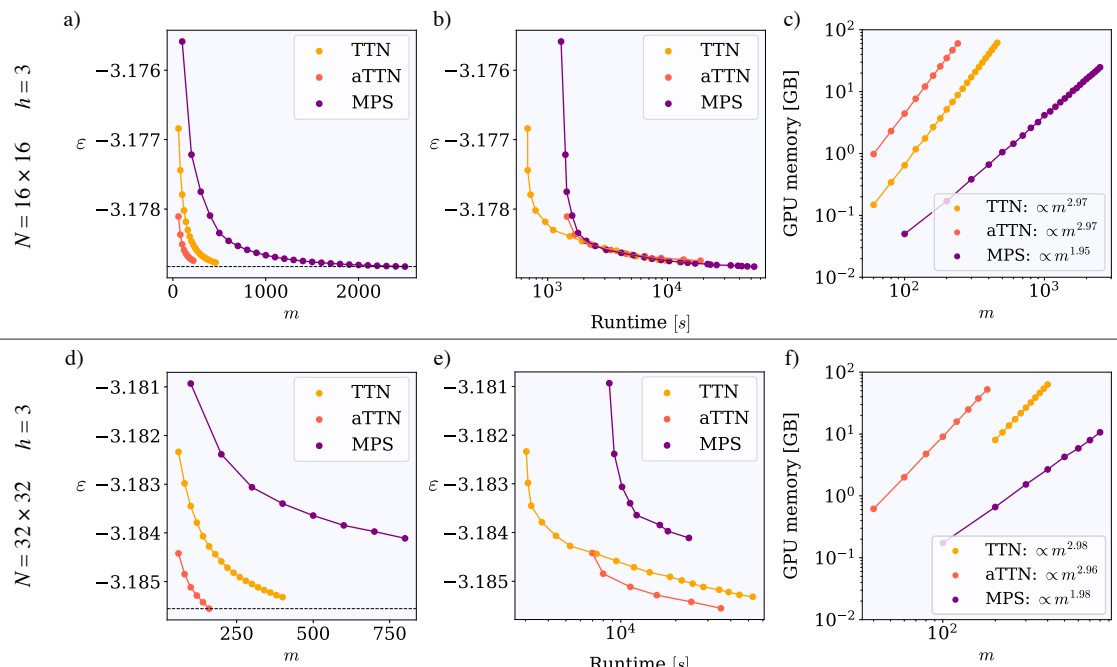

Figure 28: **Ground state search benchmark for the quantum Ising model on $L \times L$ square lattice at transverse field $h = 3$; comparison between MPS, TTN, and aTTN**. For $L = 16$: (a) ground state energy densities $\varepsilon$ as a function of bond dimension $m$ and (b) as a function of the runtime. (c) Maximal allocated GPU memory during the simulation. For $L = 32$: (d) ground state energy densities $\varepsilon$ as a function of bond dimension $m$ and (e) as a function of the runtime. (f) Maximal allocated GPU memory during the simulation.

For $L = 16$, we find that MPS performs slightly better than TTN and aTTN at the maximal reachable bond dimension $m$ (Fig. 28(a)). All three energy densities are within $10^{-5}$ difference. By comparing the energy density over runtime (Fig. 28(b)), we find very similar performance between all three ansätze. However, as we increase the lattice size to $L = 32$, a clear separation emerges, with the aTTN performing the best (Figs. 28(d) and (e)).

The underlying reason is that when mapped to a 1D tensor network, larger lattices produce more long-range interactions. These are handled better by the TTN's hierarchical structure compared to the MPS. However, at $L = 16$, the MPS is still able to compensate by reaching a larger bond dimension due to the better scaling in memory. The main factor affecting the difference between aTTN and TTN is the number of disentanglers, with a larger lattice allowing to place more of them. As discussed in Sec. 6.1.2, placing more disentanglers leads to greater energy improvement in aTTN. As a result, the tradeoff between additional cost in aTTN memory resources and gained precision shifts: for $L = 16$, the benefit of adding disentanglers does not outweigh the increase in required resources, whereas for $L = 32$, where significantly more disentanglers can be employed, the precision gain justifies the additional memory requirement.

The peak memory allocated on a GPU during the simulation (Figs. 28(c) and (f)) clearly exhibits polynomial scaling with bond dimension $m$ for all three ansatze. The scaling underlines the size of the tensors in the Lanczos algorithm for each ansätze: $\mathcal{O}(m^2)$ for MPS and $\mathcal{O}(m^3)$ for both TTN and aTTN (Fig. 19). On the GPU, we store only the isometry center tensor, Lanczos tensors, and the corresponding effective operators; thus, this behaviour precisely reproduces the theoretical expectation. We obtain the scaling coefficients by fitting the peak memory to a polynomial $A \cdot m^{\alpha}$, with the precise values obtained for $A$ and $\alpha$ shown in Table 1.

The difference between the TTN and the aTTN is in a proportionality factor $A$, with the aTTN simulations requiring roughly 6.5 times more GPU memory with respect to the TTN for $L = 16$ and 9.8 times more for $L = 32$. The increased prefactor is a consequence of the increased bond dimension of the TPO terms in the auxiliary Hamiltonian after the contraction of the disentangler layer. This contrasts with the TTN, where all Hamiltonian TPO terms maintain a bond dimension of one. Moreover, the larger prefactor for $L = 32$ with respect to $L = 16$ directly reflects the increased number of TPO terms in a single effective operator in the larger system (discussed in Sec. 4.4.2).

|  |  | MPS | TTN | aTTN |
|---|---|---|---|---|
| L = 16 | $A$ | $(5.7 \pm 0.3) \cdot 10^{-6}$ | $(7.7 \pm 0.2) \cdot 10^{-7}$ | $(5.06 \pm 0.09) \cdot 10^{-6}$ |
|  | $\alpha$ | $1.950 \pm 0.007$ | $2.966 \pm 0.005$ | $2.973 \pm 0.003$ |
| L = 32 | $A$ | $(1.9 \pm 0.1) \cdot 10^{-5}$ | $(1.13 \pm 0.03) \cdot 10^{-6}$ | $(1.10 \pm 0.04) \cdot 10^{-5}$ |
|  | $\alpha$ | $1.98 \pm 0.01$ | $2.978 \pm 0.004$ | $2.962 \pm 0.008$ |

Table 1: **Memory scaling coefficients for quantum Ising model on $L \times L$ square lattice.** Obtained scaling coefficients when fitting the values of maximal allocated GPU memory (in GB) to a polynomial $A \cdot m^{\alpha}$.

Altogether, when considering fixed computational resources, the aTTN gives an advantage for large enough lattice sizes. To further identify the Hamiltonian parameter regimes in which this advantage persists, for $L = 32$ we compare the best possible result for aTTN and TTN across different values of the external field. The best energy values are obtained at bond dimension $m = 160$ for aTTN and $m = 400$ for TTN. The resulting energies for MPS are higher than both TTN and aTTN in the entire parameter range.

We plot the difference between the best obtained energy density for the TTN and the aTTN in Fig. 29. The background colors denote which ansatz gives the best accuracy at the corresponding external field value, yellow denoting the TTN, and red denoting the aTTN. The TTN is more successful in the bulk of the phase where the state is less entangled, while the aTTN becomes advantageous when the state becomes more entangled close to the critical point. This happens because the correlations in the system grow as the critical point is approached, reaching the regime in which the high-layer TTN links become saturated. Placing the disentangler over high-layer links allows us to capture long-range correlations more accurately.

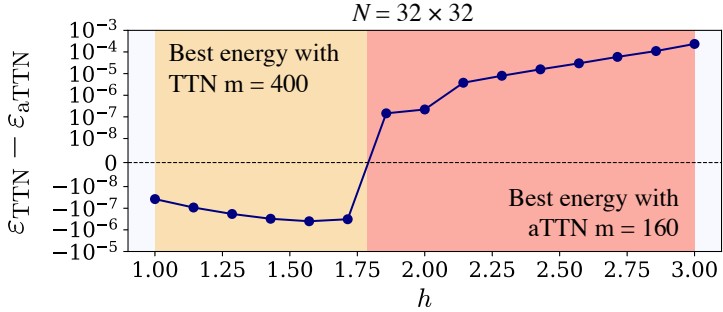

Figure 29: **The difference between the best obtained TTN and aTTN energy density with given memory resources for $L = 32$ quantum Ising model**. The difference is plotted across different external fields. Background color denotes the ansatz for which the lower energy was obtained, yellow for a TTN and red for an aTTN. The bond dimensions are $m = 400$ for TTN and $m = 160$ for aTTN.

### 6.2.2  Heisenberg model on a triangular lattice

As for the quantum Ising model, we compare the accuracy and computational cost of aTTN with TTN and MPS across different bond dimensions for the Heisenberg model (Eq. (9)) on a triangular $L \times L$ lattice. Fig. 30 shows the results for $L = 16$ and $L = 32$. This model is geometrically frustrated and is shown to be among the most challenging models for the variational ground state search [43]. We thus expect a large amount of entanglement in the ground state. The red line in Fig. 30(a) marks the lowest energy density value shown in the collection of benchmarks in Ref. [43], obtained using the recurrent neural network (RNN) wave function [44].

The best performance with given resources for $L = 16$ is again obtained by MPS. It provides the highest overall accuracy at bond dimension $m = 1200$ (Fig. 30(a)), as well as the smallest runtime at fixed accuracy (Fig. 30(b)). However, both the best performance and accuracy for $L = 32$ are obtained by TTN (Figs. 30(c) and (d)). Therefore, with given resources, the aTTN does not outperform other ansätze for this model. We attribute this to two key factors. First, because of the geometry of the lattice, the total number of interaction terms in the Heisenberg model is nine times larger than in the quantum Ising model. This means that more interaction terms will have an increased bond dimension after the disentangler layer is applied to the Hamiltonian MPO. This increases the prefactor in the aTTN's memory scaling with respect to the TTN and the MPS (recall Sec. 4.4.2), supported by the analysis of the peak GPU memory below. The largest bond dimension we could reach with the aTTN is $m = 100$. Second, the triangular lattice with nearest-neighbour interactions imposes stricter geometric constraints on the placement of disentanglers compared to the square lattice, resulting in a smaller number of viable disentangler positions. Specifically, the triangular lattice nearest-neighbour model is equivalent to a model with next-to-nearest-neighbour interactions along one diagonal on a square lattice. This limits the number of disentangler positions due to the constraint that no more than one disentangler can be assigned to a single interaction (TPO) term (Sec. 6.1.1).

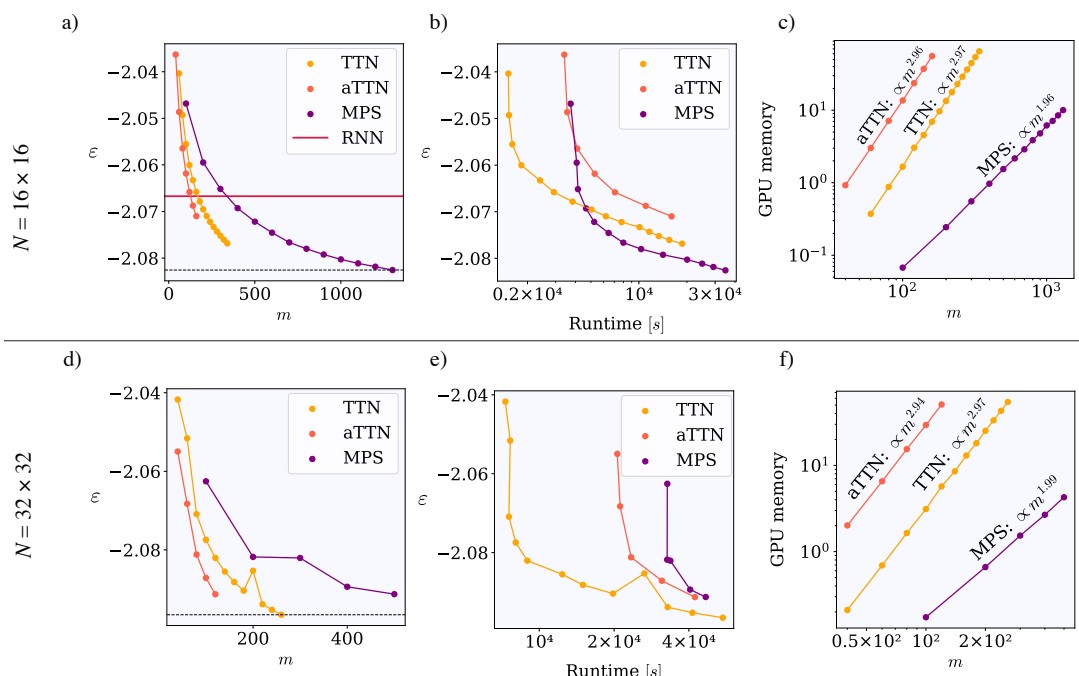

Figure 30: **Ground state search benchmark for triangular Heisenberg model on** $L \times L$ **triangular lattice; comparison between MPS, TTN and aTTN**. For $L = 16$: (a) ground state energy density as a function of bond dimensions. The red line marks, up to our knowledge, the best-known variational benchmark obtained with the recurrent neural network (RNN) wave function, taken from Refs. [43,44]. (b) Ground state energy density as a function of the corresponding runtime. (c) Maximal allocated GPU memory during the simulation. For $L = 32$: (d) ground state energy density as a function of bond dimensions and (e) as a function of corresponding runtime. (f) Maximal allocated GPU memory during the simulation.

As in the case of quantum Ising model, the peak memory allocated on a GPU during the simulation (Figs. 30(c) and (f)) scales approximately as $\mathcal{O}(m^2)$ for MPS and $\mathcal{O}(m^3)$ for both TTN and aTTN. We obtain the scalings by fitting the memory values to a polynomial $A \cdot m^\alpha$, with the precise values obtained for $A$ and $\alpha$ shown in Table 2. We find that the aTTN simulations require roughly 8.7 times more GPU memory with respect to the TTN for $L = 16$ and 10.9 times for $L = 32$. The prefactor $A$ is larger with respect to the quantum Ising model for both system sizes.

|  |  | MPS | TTN | aTTN |
|---|---|---|---|---|
| L = 16 | $A$ | $(7.8 \pm 0.4) \cdot 10^{-6}$ | $(1.93 \pm 0.07) \cdot 10^{-6}$ | $(1.675 \pm 0.0.05) \cdot 10^{-5}$ |
|  | $\alpha$ | $1.961 \pm 0.008$ | $2.973 \pm 0.007$ | $2.958 \pm 0.008$ |
| L = 32 | $A$ | $(1.8 \pm 0.2) \cdot 10^{-5}$ | $(3.6 \pm 0.2) \cdot 10^{-6}$ | $(3.9 \pm 0.2) \cdot 10^{-5}$ |
|  | $\alpha$ | $1.99 \pm 0.02$ | $2.974 \pm 0.009$ | $2.940 \pm 0.009$ |

Table 2: **Memory scaling coefficients for Heisenberg model on** $L \times L$ **triangular lattice.** Obtained scaling coefficients when fitting the values of maximal allocated GPU memory (in GB) to a polynomial $A \cdot m^\alpha$.

A reduction in the required memory of the aTTN could be achieved by compressing multiple Hamiltonian TPO which act on the same site, e.g., for the triangular Heisenberg those

would be $\sigma_x\sigma_x$, $\sigma_y\sigma_y$, and $\sigma_z\sigma_z$. This would produce TPO terms with a larger bond dimension. However, after the contraction with the disentangler layer, the compressed Hamiltonian TPO terms with fully overlapping disentanglers would result in one term with bond dimension $d^2$, instead of multiple terms with bond dimension $d^2$ as in the case without compression. We leave this for future work.

## 6.3  Disentangler optimization gain across sweeps

Finally, we tune the disentangler optimization across the DMRG sweeps. In particular, we perform $S_{\mathrm{TTN}}$ sweeps of DMRG on the TTN without the disentangler layer, and after that, start optimizing the disentangler layer in every sweep. The total number of sweeps in all simulations is 30 and the simulations were performed fully on a GPU. The benchmark results for the quantum Ising model at $h = 3$ for $32 \times 32$ lattice are shown in Fig. 31. The results show that the optimal scenario is performing the first sweep with TTN only and the rest with the disentangler layer optimization, i.e., $S_{\mathrm{TTN}} = 1$. This implies that, when starting from an already optimized TTN, the disentangler optimization is not able to escape from a local minimum.

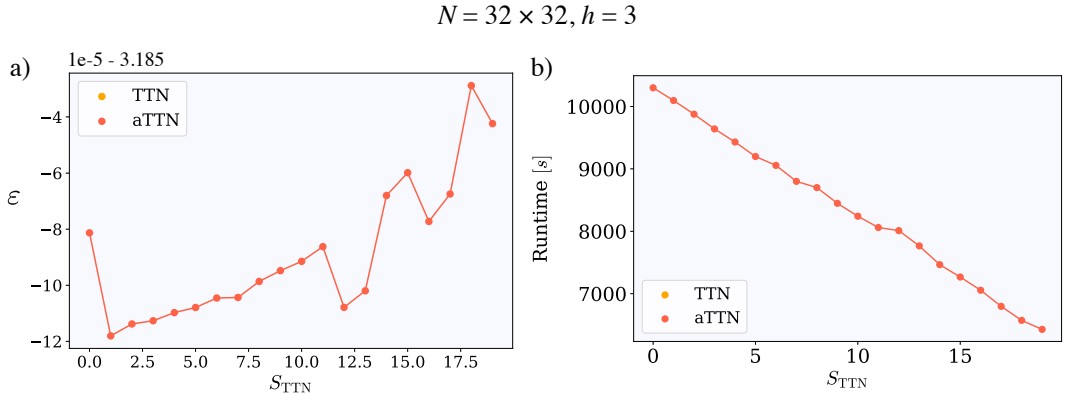

Figure 31: **Ground state search benchmark for quantum Ising model on** $32 \times 32$ **square lattice for different number of initial DMRG sweeps without the disentangler layer,** $S_{\mathrm{TTN}}$. After $S_{\mathrm{TTN}}$ sweeps, the disentangler layer is optimized in every sweep. The total number of sweeps is always 30. The plots show (a) ground state energy density $\varepsilon$ and the corresponding runtime (b) as a function of $S_{\mathrm{TTN}}$ at external field $h = 3$ and bond dimension $m = 100$.

## 7  Conclusion

We described the aTTN algorithms for the ground state search and the measurement of observables in detail. The steps presented here follow the implementation in Quantum TEA [28], an open-source tensor network software package. We benchmarked the ground state search for different hyperparameters on the quantum Ising model on a square lattice of $32 \times 32$ spins. We described the optimal strategy for the placement of disentanglers, and showed that the best accuracy is obtained when the disentangler layer optimization is performed in every sweep, starting after the first DMRG sweep on a TTN.

Furthermore, we performed large-scale ground state search simulations utilizing GPUs to compare the performance of the aTTN to the TTN and the MPS for a square lattice quantum Ising model and a triangular lattice Heisenberg model. Our results demonstrate that the aTTN consistently improves ground state energy estimates at fixed bond dimension. The largest

gains appear near the critical point, as demonstrated for the 2D quantum Ising model. To identify the parameter regimes where the aTTNs offer advantages over the MPS and the TTN in accuracy relative to computational cost, we performed the simulations across a wide range of bond dimensions, with the largest bond dimension being the largest possible given the assigned memory resources. On both the Ising and the Heisenberg models, we confirmed that the memory cost of the aTTN retains the same polynomial scaling with the bond dimension as the TTN, but with a different constant prefactor.

We find that the aTTN is advantageous for large lattices, where it is possible to place a large number of disentanglers, and close to the critical point, where the disentanglers help capture the long-range correlations. While the advantageous regime is readily in reach for the Ising model on a square lattice, we find that the aTTN does not outperform the TTN and the MPS for the Heisenberg model on the triangular lattice. The main limitation is given by the memory cost of the aTTN algorithm. This is especially apparent in the Heisenberg model, where the large number of interaction terms amplify the memory overhead post-disentangler application. We think this could be improved by using the compression of Hamiltonian terms. The regime of the utility of the aTTN can be expanded further by distributing the simulations over multiple GPUs. Finally, while up to now we have constructed the optimization scheme for finding the aTTN ground state, the next expected step is extending the aTTN algorithm scope towards the time evolution.

## Data and code availability

All the simulations are performed using the Quantum TEA libraries [28]. The benchmark datasets are available on Zenodo [45] and figures are available on the Figshare repository [46]. The simulation setup with Quantum TEA is provided within the repository [29].

## Acknowledgments

We acknowledge Flavio Baccari, Timo Felser, and Alberto Giuseppe Catalano for useful discussions. The research leading to these results has received funding from the following organizations: European Union via UNIPhD programme (Horizon 2020 under Marie Skłodowska-Curie grant agreement No. 101034319 and NextGenerationEU), Italian Research Center on HPC, Big Data and Quantum Computing (NextGenerationEU Project No. CN00000013), project Eu-RyQa (Horizon 2020), project PASQuanS2 (Quantum Technology Flagship); Italian Ministry of University and Research (MUR) via: Quantum Frontiers (the Departments of Excellence 2023-2027); German Federal Ministry of Education and Research (BMBF): project QRydDemo (the funding program quantum technologies - from basic research to market); the World Class Research Infrastructure - Quantum Computing and Simulation Center (QCSC) of Padova University; Istituto Nazionale di Fisica Nucleare (INFN): iniziativa specifica IS-QUANTUM. We acknowledge computational resources from Cineca on the Leonardo machine.

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
