# Peer review of "The Augmented Tree Tensor Network Cookbook"

_SciPost Physics Lecture Notes_

## Round 1 · Referee Report · Anonymous (Referee 1) · 2025-11-1

Strengths

1- It fills a gap in the literature about a new TN method that has sparkled curiosity and expectations in the community

2- It provides nice benchmarks according to different metrics, both conceptual (precision as a function of the bare bond dimension) and practical (precision as a function of the time-to-solution)

3- It provides direct guidance for a publicly available code, and it puts all benchmark data at disposal

4- It is nicely written and provides lots of useful illustrations (contraction schemes, disentangler placement, etc.)

Weaknesses

1- It does not touch upon the possibility of real-time dynamics, a major field of interest where TTN have already made impact... what about aTTN?

2- It does limit the benchmarks to other loop-free TN, avoiding to challenge the native 2D structures like PEPS or, at least, their isometric version (not even mentioned). Public codes are available for both, and the respective cerators would most probably be happy to provide support.

3- It does not discuss performances in periodic boundary conditions, which are arguably one of the realms where TTN still have a good advantage over other methods (including PEPS): does it extend to aTTN as well?

4- There are a number of explanations (see the alternance between MPO and TPO notation) that appear to be clumsier than needed, and whose streamlining could considerably ease the accessibility of the Lecture Notes and thus increase their impact.

Report

The Manuscript by Reinić et al. offers a practical guide for the implementation and application of augmented tree tensor networks (aTTN), which some of the Authors have introduced a few years ago and then employed in a number of recent publications. Standard binary TTN already offer considerable advantages with respect to matrix product states (MPS) when it comes to tackle two-dimensional systems with loop-free TN structures. The key idea here is to “augment” the standard binary TTN with some extra disentanglers at the level of physical degrees of freedom to better encompass the area law, while still paying an acceptable computational overhead. The performances are tested fairly thoroughly for a couple of prototypical models (like Ising in transverse field on square lattice and frustrated Heisenberg on a triangular one). Notably, practical comparisons in terms of the time-to-solution are provided. Quite honestly, the Authors recognise that the promised improvement by aTTN with respect to more traditional methods like TTN or MPS is — at the moment — only achievable for the Ising case on large lattices, due to limitations (mainly in the computational costs, but also in the deposition of the disentanglers for the triangular structure) which impairs pushing the bond dimension high enough. Nonetheless, it is very interesting to see that — conceptually — the disentangler strategy achieves its scope, with a better precision at considerably lower bond dimension than other ansätze, especially for large lattices.

The Manuscript certainly fills a gap in the literature, especially since it is accompanied by the release of an open-source implementation for all the readers and practitioners that wish for a quick application. In this sense, the Manuscript could as well have fitted into the Codebase section of SciPost.

I am very positively oriented towards publication, although some points listed below should probably be addressed before taking that step. For convenience, I divide them in three groups according to their importance. I hope the Authors will agree that, by addressing these points, the impact of their Lecture Notes could considerably increase.

——————

1) The Cookbook, as the Authors nickname their Manuscript, offers recipes for ground-state search and measurement of observables, while it does not touch about the possibility (or maybe the no-go reasons) for real-time dynamics with aTTN. Some comments in that direction seem desirable for a set of lecture notes.

2) Unfortunately, no comparison is attempted with native two-dimensional TN structures like projected entangled pair states (PEPS), whose numerical optimization is certainly more intricate but has recently experienced considerable leaps forward that promise to make them competitive, especially at large system sizes where the boundary law of entanglement kicks harder in. Incidentally, variants of PEPS restricted to isometric tensors have also been introduced, and are not mentioned in the text.

3) One crucial advantage of TTN with respect to other methods, even to PEPS (at the present day), is the possibility of dealing with periodic boundary conditions (PBC) without any extra algorithmic cost. This feature that has been exploited by some of the Authors in the past, and I assume it will hold for aTTN as well. I was surprised not to see any explicit discussion / benchmark of that in the manuscript, except for a quick mention in Sec. 3.3, where it is anyway not thematised. Does it in anyway affect the benchmark of Fig. 30 against neural quantum states, which are often more precise on PBC, as far as I know as an outsider?

——————

i) Minor annotation: the polynomial number of coefficients of a TN does not automatically imply a polynomial scaling of the underlying operations and algorithms, see indeed the case of PEPS where the exact contraction would be exponentially hard, and only (well-controlled) approximate contractions are polynomially feasible. Maybe the statement in the second sentence of the Introduction could be made a bit more precise.

ii) When comparing the scaling of computational costs (and/or memory) with the bond dimension across different TN methods, one should probably recall that the numbers to be plugged in might be considerably different. E.g., MERA promise to have a constant and very moderate bond dimension even for critical systems (though their algorithms are of little practical use, in the end), and PEPS bond dimensions are typically very small (in the range up to ten, or so), since they rely on their structure to enhance the descriptive power (a back-of-the-envelope estimate tells \chi_{PEPS}^L ~ m_{MPS}).

iii) I was a bit surprised not to see a remark about the boundary-law being captured “on average” by the TTN structure, as some of the Authors have shown in the past. This certainly goes to the merit of TTN and explain their success in dealing with many two-dimensional problems despite their loop-free structure. I somehow found the statement “However, the TTN architecture does not capture the area law in two dimensions [23,24]” even too severe.

iv) Conversely, the statement “[…] the augmented tree tensor network (aTTN) ansatz […] geometry enables capturing the entanglement area law in any dimension [25]” seems a bit overstretched. From what I understand in Fig. 25 (by the way, why so late in the Manuscript?), the insertion of disentanglers helps to considerably improve the number of tensor legs cut by a bipartition, but still do not ramp it up to full-glory boundary scaling, not? Most probably, I am overlooking something that was explained in Ref. [25]: if this is the case, I would suggest to keep the cookbook self-contained and to report here the proof for the claim. What puzzles me is that an area-law ansatz should allow for very high predictive power at very small bond-dimensions (independently of system size), which does not seem the empirical case here — at fixed m, the aTTN is more expressive, but we are still talking of m~100 or more (and scaling with system size).

v) At the end of Sec. 2, there is a rather mysterious statement about different approaches reported in previous references by the same group: Could the Authors explain in a nutshell what are the advantages/disadvantages of the two, which lead them to write the Cookbook according to a specific one? A self-contained presentation is an important feature of Lecture Notes.

vi) When looking at Fig. 10 and surrounding text, and I look back to the Anthology by some of the Authors (where TTN were discussed), I have the impression that the construction of environments could be even more compactly described in terms of (a subset of) the three Hamiltonian environments (one per leg) of the isometries along the percolation path of the disentangler (up to the anchor tensor, see Fig. 12). Would it be a fair description that helps all those already acquainted with TTN and possibly having a TTN algorithm at disposal (an increasing number of people)?

vii) I must confess that I do not understand the statement “Recall that we only have to do this contraction along the path, as the rest of the network remains unitary and thus cancels out”, since to me it seems that this is neglecting the important case of Hamiltonian terms that partially operate on the disentangler sites and partially outside… so that, in general, their second part makes the rest of the network not isometric/unitary. In other words, I am confused by the (absent?) distinction between the disentangler path defined in Fig.12 and the path between the sites on which each Hamiltonian term is acting. Maybe everything simplifies considerably with the suggestion at point vi)? Could the Author comment on that?

viii) Another thing that I find very confusing, and not so appropriate for a Lecture Note, is the oscillation between MPO and TPO notation in the description of the algorithm. As the Authors correctly state, the MPO bond dimension t for a 2d lattice would be very large and scaling with system size. This is precisely the reason why in 2D TTN one never goes through that pain but rather through the smaller one of bookkeeping the TPO terms explicitly. Therefore, I do not understand why there are discussions about the scaling ~t^4 of the memory requirements or later ~t^3 of the contraction costs, then toned down by observations like “in most cases, t=1”. Can the Authors please streamline the presentation, in particular around Fig. 19 where the cost estimates seem overshot? Is there really some case where one needs all the TPO legs to be t-dimensional and to close around with PBC?

ix) Looking at Fig. 17, I wonder why — at least in panel a), but probably also in b) — the proposed contraction / decomposition sequence should be the optimal one. If am not terribly mistaken: on one hand, its cost scales like ~(2 t d^6 + …) due to both the contraction from second to third row and the QR from third to fourth row; on the other hand, contracting directly the disentangler, its conjugate and the two TPO tensors together to a single object of dimensions {d,d,d,d} should cost ~(2 d^6 + t d^4), and then a QR directly into the desired final form could be performed with cost ~(d^6). This seems to gain a factor ~t in the computational costs, in case this is a relevant addend to the full costs of the algorithm. However, I am not sure the latter is the case, so that my observation might be not practically relevant. I would love to see a comment by the Authors here.

x) To better assess the quality of the comparisons, it would be interesting to know whether symmetries (like Z_2 for Ising and, most importantly, SU(2) or at least U(1) for Heisenberg) have been exploited (and how) in the different algorithms. The chosen benchmarks should allow for that, or am I mistaken?

————

a) Minor: in panel 3 of Fig. 15 there should be no black dots for the uppermost tensor (and its conjugate), like there are not in the rearranged panel 4 — if I am not mistaken.

b) Minor: why do not all contraction/decomposition schemes report also the scalings of the operations like nicely done in Fig. 16 & 19?

c) Minor: in Figs. 17-18, it would be probably helpful to indicate with a different color the tensors that get changed during the procedure.

d) Minor: How does the Hilbert curve of Fig.25 differ from the alternate renormalization of pairs of neighbours along x and y direction that was presented in the first 2D TTN papers by the same group? Are they somehow equivalent, or is there some crucial difference?

e) Minor: In Fig. 23 it would be nice to graphically indicate which disentangler configurations are actually forbidden — should be only the second one, right?)

f) Minor: Some better ticking of the axes in Fig. 28 could allow to more directly read the bond dimensions at which TTN and aTTN stop.

g) Minor: for the final version of the Manuscript, it would be obviously nice to get the TTN at m=200 in Fig. 30 better converged, so that no strange bump is present :-)

h) Minor: In the conclusion, the benchmark with Heisenberg model on triangular lattice is not mentioned.

Recommendation

Ask for minor revision

---

## Editorial Decision

awaiting_resubmission